# Cell-autonomous *Toxoplasma* killing program requires Irgm2 but not its microbe vacuolar localization

Ariel Pradipta[1,2], Miwa Sasai[1,2], Kou Motani[4], Ji Su Ma[1], Youngae Lee[2], Hidetaka Kosako[4], Masahiro Yamamoto[1,2,3]

Interferon-inducible GTPases, such as immunity-related GTPases (IRGs) and guanylate-binding proteins (GBPs), are essential for cell-autonomous immunity against a wide variety of intracellular pathogens including *Toxoplasma*. IRGs comprise regulatory and effector subfamily proteins. Regulatory IRGs Irgm1 and Irgm3 play important roles in anti-*Toxoplasma* immunity by globally controlling effector IRGs and GBPs. There is a remaining regulatory IRG, called Irgm2, which highly accumulates on parasitophorous vacuole membranes (PVMs). Very little is known about the mechanism of the unique localization on *Toxoplasma* PVMs. Here, we show that Irgm2 is important to control parasite killing through recruitment of Gbp1 and Irgb6, which does not require Irgm2 localization at *Toxoplasma* PVMs. Ubiquitination of Irgm2 in the cytosol, but not at the PVM, is also important for parasite killing through recruitment of Gbp1 to the PVM. Conversely, PVM ubiquitination and p62/Sqstm1 loading at later time points post-*Toxoplasma* infection require Irgm2 localization at the PVM. Irgm2-deficient mice are highly susceptible to *Toxoplasma* infection. Taken together, these data indicate that Irgm2 selectively controls accumulation of anti-*Toxoplasma* effectors to the vacuole in a manner dependent or independent on Irgm2 localization at the *Toxoplasma* PVM, which mediates parasite killing.

## Introduction

Pathogen infection activates a series of immune responses in healthy mammalian hosts. Innate immunity initially detects pathogen-derived components through germline-encoded receptors, such as Toll-like receptors, which induces a proinflammatory cytokine called interleukin-12 (Hunter & Remington, 1995; Yarovinsky & Sher, 2006). This cytokine subsequently stimulates the acquired type I immune response through which naïve CD4+ and CD8+ T cells differentiate into pathogen-derived antigen-specific Th1 cells and cytotoxic T cells with the help of antigen-presenting cells (Gazzinelli et al, 1991). These T-cell subsets together with natural killer cells robustly produce IFN-γ to activate cell-autonomous immunity that eliminates various vacuolar pathogens,

including a major human and animal protozoan *Toxoplasma gondii* (Sturge & Yarovinsky, 2014).

IFN-γ stimulates production of hundreds of IFN-γ–stimulated proteins such as indole 2,3-deoxygenase (IDO), inducible nitric oxide synthase (iNOS), and IFN-inducible GTPases. In the infected cells, IDO and iNOS inhibit *T. gondii* proliferation by depleting tryptophan and arginine, respectively (Hunter & Sibley, 2012). IFN-inducible GTPases, such as p47 immunity-related GTPases (IRGs) and p65 guanylate-binding proteins (GBPs), localize at membranes of *T. gondii* parasitophorous vacuole (PV) in which the parasite sequesters host nutrients and proliferates efficiently (Kim et al, 2012). Accumulation of IRGs and GBPs eventually disrupts the *T. gondii* parasitophorous vacuole membrane (PVM) to kill the pathogen (Martens & Howard, 2006; Saeij & Frickel, 2017). In mice, the IRG protein family comprises three regulator proteins (Irgm1, Irgm2, and Irgm3) and >20 effector proteins (Bekpen et al, 2005; Pilla-Moffett et al, 2016). Effector IRG proteins harbor a universally conserved GX₄GKS sequence in the nucleotide-binding motif (G1), which enables binding to both GDP and GTP (Uthaiah et al, 2003; Bekpen et al, 2005). Conversely, the G1 motif of regulator IRG proteins possess a GX₄GMS sequence that mainly binds to GDP by which effector IRGs are maintained in an inactive state and their activation is prevented (Bekpen et al, 2005; Hunn et al, 2008). Such negative regulation might be important for protection of host endomembranes, because Irgm1 and Irgm3 localize at the host Golgi apparatus and ER, respectively (Martens et al, 2004; Hunn et al, 2008; Haldar et al, 2013). Lack of Irgm1 and Irgm3 results in the formation of nucleotide-dependent cytoplasmic aggregates that are caused by premature GTP binding and activation of effector IRGs, which leads to severely impaired loading of effector IRGs and GBPs on the *T. gondii* PVM and dismantling cell-intrinsic immunity (Taylor et al, 2000; Maric-Biresev et al, 2016). Thus, IRG and GBP-dependent cell-intrinsic immunity against vacuolar pathogens is well known to be globally controlled by Irgm1 and Irgm3. In addition, the localization of IRGs and GBPs is regulated by autophagy-related proteins (ATGs) in a manner independent of autophagy (Zhao et al, 2008; Yamamoto et al, 2012; Ohshima et al, 2014; Park et al, 2016; Sasai et al, 2017). The rest of a member of a regulator IRG protein is

[1]Department of Immunoparasitology, Research Institute for Microbial Diseases, Osaka University, Osaka, Japan    [2]Laboratory of Immunoparasitology, WPI Immunology Frontier Research Center, Osaka University, Osaka, Japan    [3]Division of Microbiology and Immunology, Center for Infectious Disease Education and Research, Osaka University, Osaka, Japan    [4]Division of Cell Signaling, Fujii Memorial Institute of Medical Sciences, Tokushima University, Tokushima, Japan

Correspondence: myamamoto@biken.osaka-u.ac.jp

Irgm2, which localizes to the *T. gondii* PVM as well as the host Golgi apparatus (Hunn et al, 2008). However, the physiological role of Irgm2 and particularly the molecular mechanism of *T. gondii* PVM localization remain uncertain.

Effector IRG proteins, such as Irga6, Irgb6, and Irgb10, accumulate on the *T. gondii* PVM in a hierarchical manner (Khaminets et al, 2010). Irgb6 initially detects the *T. gondii* PVM via its phospholipid binding and leads to subsequent loading of Irga6 and Irgb10 (Lee et al, 2020), recruitment of GBPs that comprise 11 members in mice (Kresse et al, 2008). Some effector IRG proteins mutually control the localization of GBPs. Irgb6 controls accumulation of Gbp1 on the PVM and vice versa (Selleck et al, 2013) and Gbp2 regulates Irga6 loading on the PVM (Ohshima et al, 2015). Accumulation of effector IRGs and GBPs in turn damages the PVM (Martens et al, 2005; Kravets et al, 2016), which leads to PVM ubiquitination, followed by p62/Sqstm1 coating (Haldar et al, 2015; Lee et al, 2015). One study has demonstrated that ubiquitination of *T. gondii* PVM is important for parasite killing (Haldar et al, 2015). A subsequent recent study has demonstrated roles of TRAF6 and TRAF2 in PVM ubiquitination and TRAF6-dependent killing of *T. gondii* in IFN-γ–primed mouse fibroblasts (Mukhopadhyay et al, 2020). Conversely, another study has shown that ubiquitination and p62 coating is dispensable for *T. gondii* killing (Lee et al, 2015). Thus, the biological significance of IFN-γ–inducible *T. gondii* PVM ubiquitination is a matter of debate. Moreover, the ubiquitin substrate(s) on *T. gondii* PVM is unclear.

Here, we aimed to determine the role of Irgm2 in cell-intrinsic immune responses to *T. gondii*. We found that Irgm2 specifically participates in Irgb6 and Gbp1-mediated parasite killing by regulating their accumulation on the *T. gondii* PVM. Furthermore, a cysteine residue in the C-terminus of Irgm2 determines its localization on the *T. gondii* PVM, which is important for prolonged loading of ubiquitin and p62 on the PVM but not for Irgb6- and Gbp1-mediated parasite killing. Moreover, we found that ubiquitinated Irgm2 is required for Gbp1-mediated *T. gondii* killing but may not localize at the PVM. Finally, we found that Irgm2-deficient mice are highly susceptible to *T. gondii* infection. Collectively, these data demonstrate that IFN-γ–induced cell-autonomous immunity against *T. gondii* is regulated in a manner dependent or independent on Irgm2 PVM localization.

# Results

### Irgm2-deficient cells are defective for recruitment of *T. gondii* clearance-related effectors

To assess the role of Irgm2 in anti–*T. gondii* immune responses, we generated Irgm2-deficient mice by CRISPR/Cas9 genome editing (Fig S1A and B). Anti-Irgm2 did not detect Irgm2 proteins in MEFs from Irgm2-deficient mice (Fig S1A and B). We first obtained BMDMs, BMDCs, and MEFs from wild-type and Irgm2-deficient mice, and compared the IFN-γ–induced reduction of *T. gondii* parasite numbers in the cells (Fig 1A). We found that Irgm2–deficient BMDMs, BMDCs, and MEFs were defective in IFN-γ–induced reduction of *T. gondii* numbers in comparison with wild-type cells (Fig 1A). However, the magnitude of the defect in single Irgm2-deficient cells was not as severe as Irgm1/Irgm3 double KO (DKO) or Irgm1/Irgm2/Irgm3 triple knockout (TKO) cells (Figs 1A and S1C), which indicates that Irgm2 as

well as Irgm1 and Irgm3 contributes to the IFN-γ–mediated cell-autonomous response to *T. gondii*. Recruitment of effector IRGs and GBPs has been extensively studied as important host cell-intrinsic events for *T. gondii* killing. We compared localization of effector IRGs, such as Irga6 and Irgb6, and GBPs, such as Gbp1 and Gbp2, on the *T. gondii* PVM in wild-type and Irgm2-deficient cells (Fig 1B and C). We confirmed that IRG and GBP effector proteins were expressed at comparable levels in wild-type and Irgm2-deficient cells (Fig 1D). Taken together, these results demonstrate that Irgm2 deficiency impairs *T. gondii* killing activity with selectively decreases recruitment of Irgb6 and Gbp1 on the PVM.

### The GMS configuration of the Irgm2 GTPase domain is essential for *T. gondii* clearance activity

We next explored the molecular mechanisms by which Irgm2 regulated the IFN-γ–induced cell-autonomous immune response. Irgm2 possesses an N-terminal GTPase domain (Figs 2A and S1D). It has been previously shown that the GKS configuration in the N-terminal GTPases of Irga6 and Irgb6 is important for their recruitment function to the *T. gondii* PVM (Hunn et al, 2008; Papic et al, 2008; Lee et al, 2020). Irgm2 harbors a GMS sequence in its GTPase domain, in which the mutated methionine is a part of the unconventional GMS P-loop sequence of immunity-related GTPase family M (IRGM) proteins (Bekpen et al, 2005). Therefore, we examined the role of the Irgm2 GTPase domain. A point mutant, in which the methionine at position 77 in the GTPase domain of Irgm2 was substituted with alanine (Irgm2 M77A), may disrupt nucleotide (GDP) binding and hence cause non-functionality (Hunn et al, 2008). We reconstituted wild-type and M77A Irgm2 in Irgm2-deficient MEFs for IFN-γ–induced reduction of *T. gondii* numbers and recruitment of Irgb6 and Gbp1 (Fig 2B and C). We reconstituted Irgm2-deficient MEFs with wild-type Irgm2 and the M77A mutant, and confirmed similar expression (Fig 2B). When IFN-γ–induced killing activity was examined, Irgm2-deficient MEFs reconstituted with wild-type Irgm2 were able to recover the killing activity (Fig 2C). In sharp contrast, Irgm2 KO MEFs that expressed the M77A mutant were not able to restore this killing activity (Fig 2C). Furthermore, reconstitution of wild-type Irgm2 in Irgm2-deficient MEFs recovered recruitment of Gbp1 and Irgb6, whereas that of the M77A mutant did not (Fig 2D–F). Wild-type Irgm2, but not the M77A mutant, was found on the *T. gondii* PVM (Fig 2D and G) as described previously (Martens & Howard, 2006; Zhao et al, 2010). Taken together, these results demonstrate that the GMS configuration of Irgm2 is essential for the cell-autonomous immune function of Irgm2.

### Regulatory IRGs and ATG proteins are dispensable for Irgm2 localization on the *T. gondii* PVM

Next, we examined the regulatory mechanism of Irgm2 localization on the *T. gondii* PVM. As reported previously (Haldar et al, 2013), localization of Irgb6 and Gbp1 was severely impaired in cells that lacked Irgm1 and Irgm3 (Fig S2A–D). Surprisingly, the localization of Irgm2 was normal in Irgm1/Irgm3 double-deficient cells (Fig 3A and B), which suggests that these regulatory IRGs do not control Irgm2 localization on the *T. gondii* PVM. In addition to regulatory IRGs, autophagy proteins, such as Atg3 and Atg7, are essential for correct targeting of effector IRGs to the *T. gondii* PVM (Choi et al, 2014; Ohshima et al, 2014; Sasai et al, 2017). However, cells that lacked Atg3

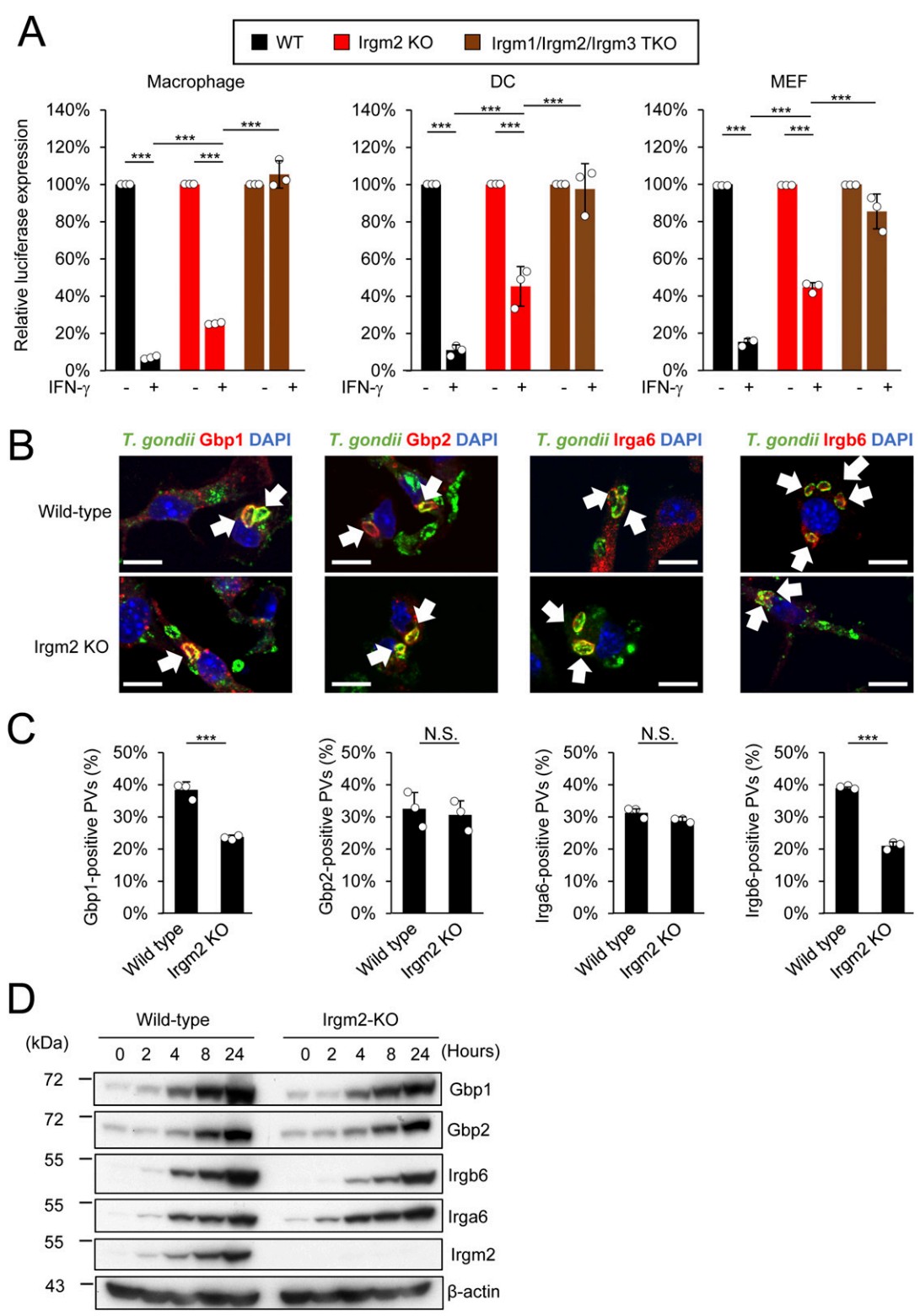

**Figure 1. Irgm2 selectively controls IFN-inducible GTPase-dependent cell-autonomous immunity.**
**(A)** *Toxoplasma gondii* survival rate in BMDM, BMD-DC, and MEF with IFN-γ stimulation relative to those without IFN-γ treatment by luciferase analysis at 24 h post infection. **(B)** Confocal microscopy images to show the localization of various effectors (red) including Gbp1, Gbp2, Irga6, and Irgb6 to *T. gondii* parasitophorous vacuole (green) and DAPI (blue) at 4 h post infection in IFN-γ treated MEFs indicated on the sides. **(C)** Recruitment percentages of Gbp1, Gbp2, Irga6, and Irgb6. **(D)** Western blot image of indicated protein expression in WT and Irgm2 KO MEFs at indicated hours post IFN-γ activation. All graphs show the mean ± SEM in three independent experiments. All images are representative of three independent experiments. N.D., not detected; *P < 0.05, **P < 0.01, ***P < 0.001. Difference in *T. gondii* inhibition activity between IFN-γ–activated versus nonactivated was subjected to two-way ANOVA, with Tukey's multiple comparisons test to analyze the difference between genotypes. Effector recruitment comparison between only two genotypes analyzed with Mann–Whitney test. White arrows indicate recruitment of effector on *T. gondii* parasitophorous vacuole. Scale bars on microscope images represent 10 μm.

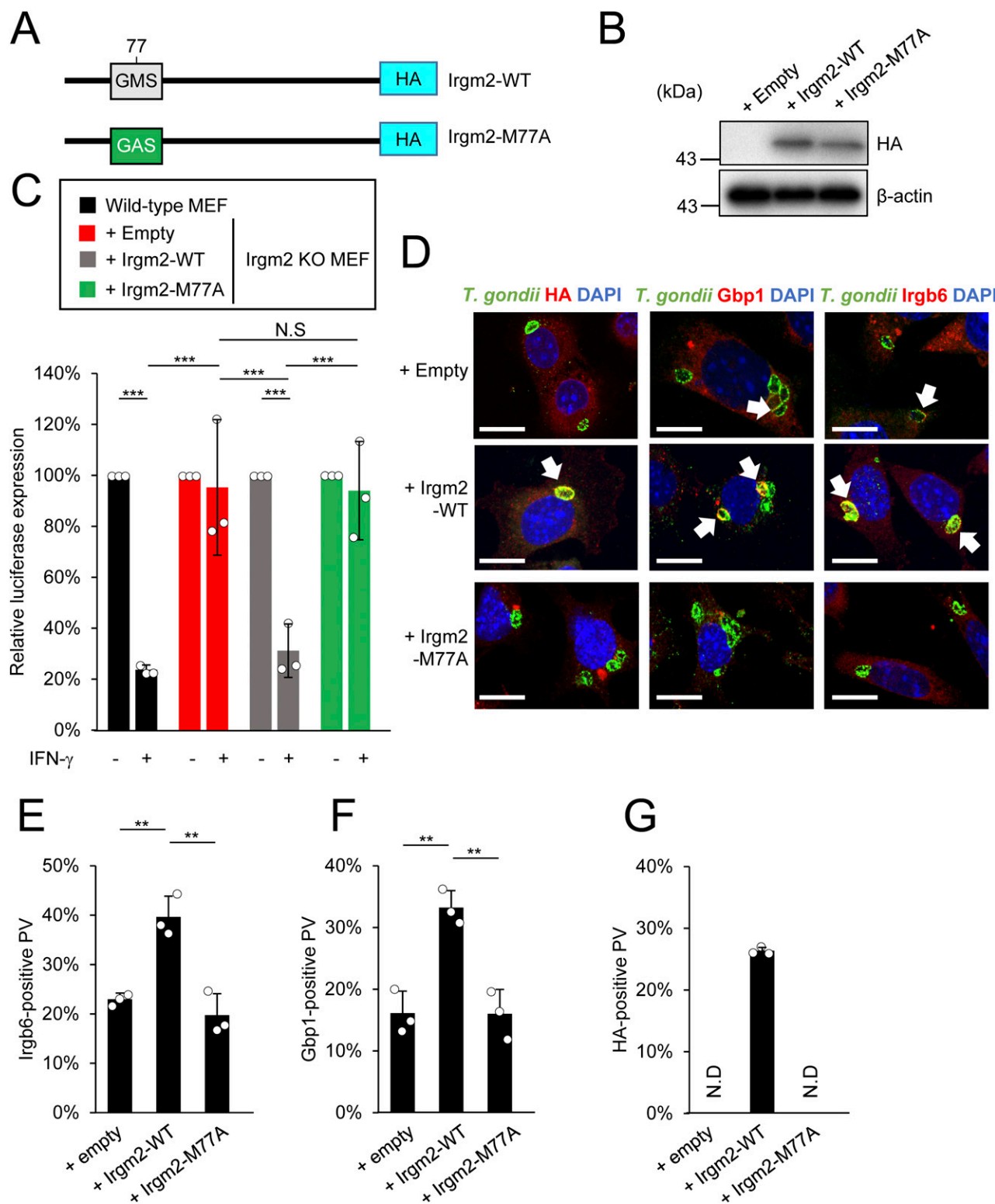

**Figure 2. The guanine nucleotide binding moiety is essential for Irgm2-dependent anti–*Toxoplasma gondii* programs.**
**(A)** Schematic overview of the substitution mutation site on the Irgm2 sequence. **(B)** Western blot image to detect stably expressed Irgm2 protein after retroviral transfection and puromycin selection. **(C)** *T. gondii* survival rate in the indicated Irgm2 reconstitution in Irgm2 KO MEFs with IFN-γ stimulation relative to those without IFN-γ treatment by luciferase analysis at 24 h post infection. **(D)** Confocal microscope images to show the localization of Irgm2-HA, Gbp1, and Irgb6 (red) to *T. gondii* parasitophorous vacuole (green), and DAPI (blue) at 4 h post infection in IFN-γ–treated Irgm2-KO MEFs reconstituted with Irgm2-WT and Irgm2-M77A. **(E, F, G)** Recruitment percentages of Irgb6 (E), Gbp1 (F), HA (G). All graphs show the mean ± SEM in three independent experiments. All images are representative of three independent experiments. N.D., not detected; *P < 0.05, **P < 0.01, ***P < 0.001. Difference in *T. gondii* inhibition activity between IFN-γ–activated versus nonactivated was subjected to two-way ANOVA, with Tukey's multiple comparisons test to analyze the difference between genotypes. Effector recruitment comparison between genotypes applied one-way ANOVA (Tukey's multiple comparisons test). White arrows indicate recruitment of effector on *T. gondii* parasitophorous vacuole. Scale bars on microscope images represent 10 μm.

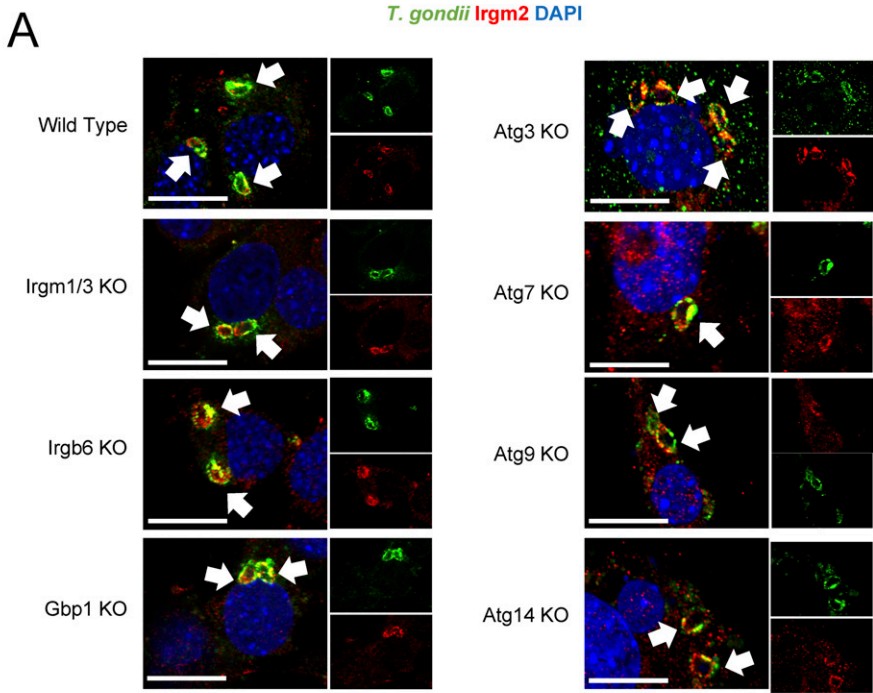

T. gondii Irgm2 DAPI

A

Wild Type

Irgm1/3 KO

Irgb6 KO

Gbp1 KO

Atg3 KO

Atg7 KO

Atg9 KO

Atg14 KO

**Figure 3. Irgm2 recruitment to *Toxoplasma gondii* parasitophorous vacuole (PV) is independent of Irgm1/Irgm3 and Atg proteins.**

**(A)** Confocal microscope images to show the localization of Irgm2 (red) to *T. gondii* PV (green) and DAPI (blue) at 4 h post infection in IFN-γ treated indicated MEFs. **(B)** Percentages of endogenous Irgm2 recruitment to *T. gondii* PV in indicated cells are shown. All graphs show the mean ± SEM in three independent experiments. All images are representative of three independent experiments. N.D., not detected. White arrows indicate recruitment of effector on *T. gondii* PV. Scale bars on microscope images represent 10 *μm*. All graphs show the mean ± SEM in three independent experiments. All images are representative of three independent experiments. N.D., not detected; *P < 0.05, **P < 0.01, ***P < 0.001. Effector recruitment comparison between genotypes applied one-way ANOVA (Tukey's multiple comparisons test). White arrows indicate recruitment of effector on *T. gondii* PV. Scale bars on microscope images represent 10 *μm*.

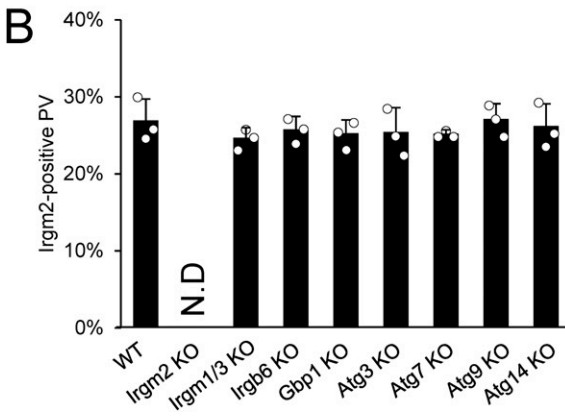

B

and Atg7 as well as Atg9 or Atg14 displayed normal Irgm2 recruitment of the *T. gondii* PVM (Fig 3A and B), which indicates that Atg proteins are dispensable for Irgm2 localization on the *T. gondii* PVM. GBPs and effector IRGs mutually regulate localization of each other (Yamamoto et al, 2012; Selleck et al, 2013; Ohshima et al, 2015; Saeij & Frickel, 2017; Lee et al, 2020). Although localization of Irgb6 or Gbp1 on the *T. gondii* PVM was affected by Gbp1 or Irgb6 deficiency (Fig S2A–D), Irgm2 localization was normal in Irgb6- or Gbp1-deficient cells (Fig 3A and B), which demonstrated that regulatory IRGs and Atg proteins are not involved in Irgm2 localization on the *T. gondii* PVM.

### A cysteine in the C terminus of Irgm2 controls localization on the *T. gondii* PVM

We further examined the molecular mechanism by which Irgm2 localizes at the *T. gondii* PVM. Irgb6, Irgb10, and Irgm1 possess an amphipathic helix in their C terminus called αK, which directs binding to the target membrane (Martens et al, 2004; Tiwari et al, 2009; Man et al, 2016; Lee et al, 2020). Localization of Irgm1, which is most homologous to Irgm2, at the host mitochondrial membrane was abolished by introducing mutations in a tight cluster of the cysteine near the C terminus of Irgm1, which is immediately adjacent to the αK region (Henry et al, 2014). When we searched for such potential cysteine residues in the corresponding C terminus of Irgm2, we found a cluster of cysteines (aa357 and aa358) adjacent to the αK (aa336–aa353) (Figs 4A and S1D). To investigate whether the cysteine residues are involved in Irgm2 targeting the *T. gondii* PVM, we generated Irgm2 C357A or C358A point mutants (Fig 4A), reconstituted them in Irgm2 KO MEFs (Fig 4B), and examined their localization in *T. gondii*–infected or uninfected cells (Figs 4C and D and S3A). Although recruitment of C357A Irgm2 to the *T. gondii* PVM was comparable to that of wild-type Irgm2, it was of interest that the

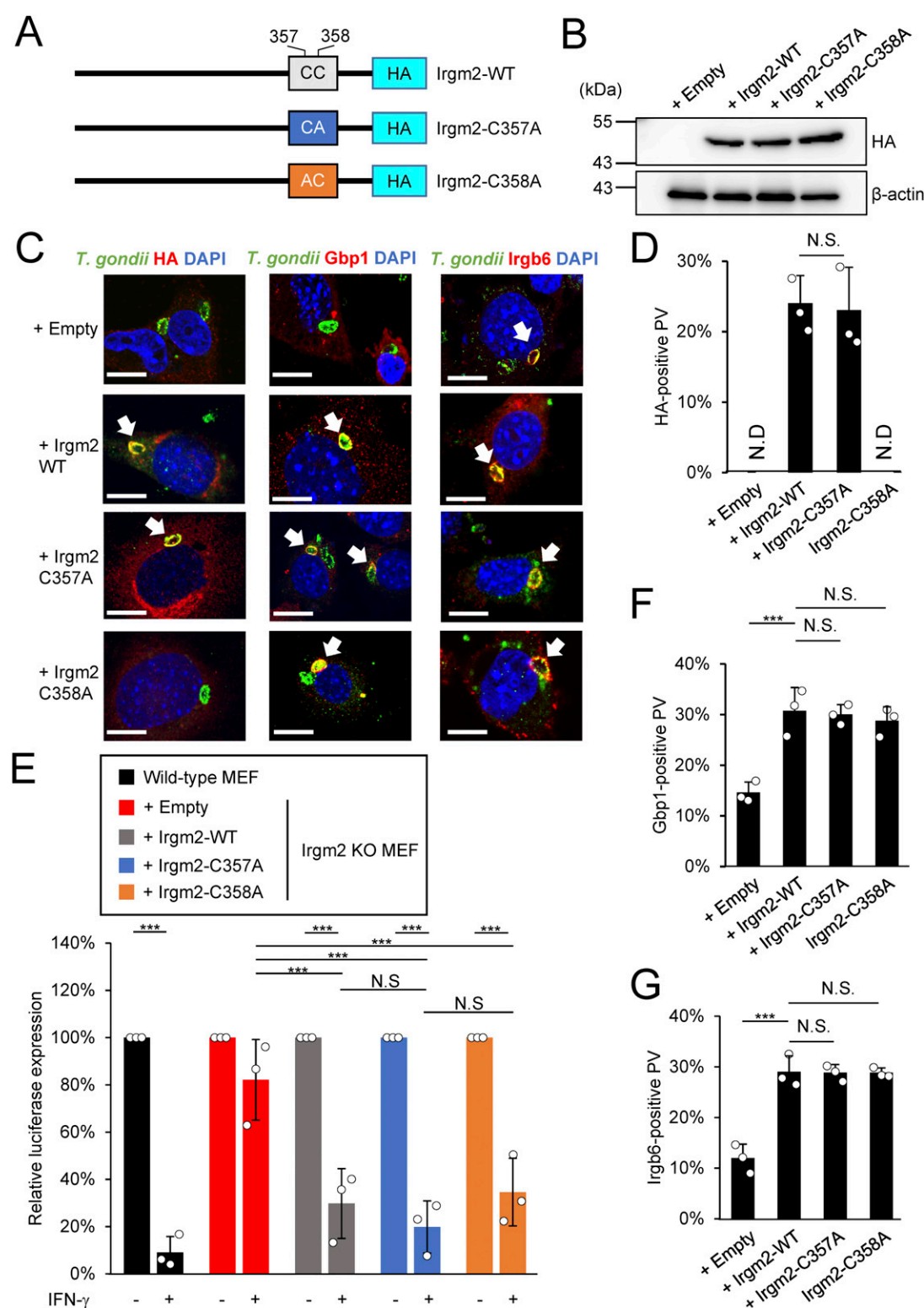

**Figure 4. The C358A mutation in the C-terminus only affects its recruitment to *Toxoplasma gondii* parasitophorous vacuole (PV) but not the killing program.**
**(A)** Schematic overview of the substitution mutation site on the Irgm2 sequence. **(B)** Western blot image to detect stably expressed Irgm2 protein after retroviral transfection and puromycin selection. **(C)** Confocal microscope images to show the localization of Irgm2-HA, Gbp1, and Irgb6 (red) to *T. gondii* PV (green), and DAPI (blue) at 4 h post infection in IFN-γ–treated Irgm2-KO MEFs reconstituted with indicated Irgm2. **(D, F, G)** Recruitment percentages of Irgm2-HA (D), Gbp1 (F), and Irgb6 (G). **(E)** *T. gondii* survival rate in the indicated Irgm2 reconstitution in Irgm2 KO MEFs with IFN-γ stimulation relative to those without IFN-γ treatment by luciferase analysis at 24 h post infection. All graphs show the mean ± SEM in three independent experiments. All images are representative of three independent experiments. N.D., not

C358A Irgm2 mutant did not accumulate on the *T. gondii* PVM (Fig 4C and D), which indicates that the Cys358 of Irgm2 is a determinant for localization on the *T. gondii* PVM. When localization of Irgm2 wild-type, C358A, and M77A mutants was tested in uninfected cells, the wild-type and M77A Irgm2 mutant were detected at the Golgi apparatus, whereas the C358A mutant was not (Fig S3A), which suggests that Cys358 is responsible for Irgm2 localization at the Golgi apparatus in uninfected cells. Next, we assessed whether Irgm2 localization on *T. gondii* is involved in IFN-γ–induced parasite clearance (Fig 4E). Surprisingly, reconstitution of the C358A mutant in Irgm2-deficient cells fully restored IFN-γ–induced *T. gondii* killing activity in a manner similar to that of wild-type Irgm2 (Fig 4E). Consistently, recruitment of Irgb6 and Gbp1 to the *T. gondii* PVM was recovered by reintroduction of the C358A mutant well as wild-type Irgm2 into Irgm2-deficient cells (Fig 4C, F, and G). Taken together, these data suggest that Irgm2 localization on the *T. gondii* PVM in infected cells as well as its accumulation at the Golgi apparatus in uninfected cells are determined by Cys358 at the C terminus, whereas it is not linked to Irgb6/Gbp1-mediated killing activity.

### Irgm2 localization on the *T. gondii* PVM is important for prolonged accumulation of p62 and ubiquitin

Recruitment of p62, ubiquitin, Irgb6, and Gbp1 to the *T. gondii* PVM was compared between wild-type Irgm2- or the C358A-reconstituted or empty vector-transduced Irgm2-deficient cells in time-dependent manners (Fig 5A–D). When we compared ubiquitin loading and recruitment of Gbp1, Irgb6 and p62 among wild-type, Irgm2-deficient, and Irgm1/Irgm3 DKO cells, we found that ubiquitin loading and recruitment of these effectors were greatly reduced in Irgm1/Irgm3 DKO cells as reported previously (Lee et al, 2015). Compared with Irgm1/Irgm3 DKO cells, Irgm2-deficient cells were partially defective for loading of these effectors (Fig S3B). Despite no difference at 4 h postinfection, it was of note that the C358A Irgm2 mutant-reconstituted or empty vector-transduced Irgm2-deficient cells showed more rapid reduction in recruitment of p62 and ubiquitin at later time points (6 and 8 h postinfection) than wild-type Irgm2-reconstituted cells (Fig 5A and B). Conversely, there was no significant difference in recovery of Irgb6 and Gbp1 recruitment to the *T. gondii* PVM between wild-type Irgm2- and C358A mutant-reconstituted cells (Fig 5C and D), which suggests that Irgm2 localization on the *T. gondii* PVM is important for prolonged recruitment of p62 and ubiquitin on parasites. Taken together, these results demonstrate that Irgm2 localization on the *T. gondii* PVM is dispensable for parasite killing but indispensable for prolonged accumulation of p62 and ubiquitin on the PVM.

### Ubiquitination of Irgm2 in the cytosol is important for Gbp1 recruitment to the *T. gondii* PVM

Next, we searched for potential sites of protein modifications, such as phosphorylation and ubiquitination, in Irgm2. We subjected Spot-tagged Irgm2 to mass spectrometric analyses and found that

several lysine residues might be ubiquitinated (Figs 6A, S1D, and S4). When Flag-tagged wild-type Irgm2 was reconstituted in Irgm2-deficient cells that expressed 3×HA–tagged ubiquitin, the immune-precipitated Flag-tagged wild-type or C358A Irgm2 was heavily ubiquitinated, whereas the immune-precipitated KA mutant, in which all of the potential lysine residues were substituted to alanines, did not yield such ubiquitin smears (Fig 6B), which suggests that the lysine residues of Irgm2 are ubiquitination sites. Next, we assessed the significance of Irgm2 ubiquitination in IFN-γ–induced anti–*T. gondii* responses. Notably, reconstitution of the Irgm2 KA mutant in Irgm2-deficient cells only partially recovered parasite killing in comparison with that of wild-type Irgm2 (Fig 6C). Recruitment of Gbp1 and Irgb6 was evaluated in the reconstituted cells (Fig 6D–F). Accumulation of Irgb6 was comparable between wild-type Irgm2- and KA mutant-reconstituted cells (Fig 6D and F). Conversely, KA Irgm2-reconstituted cells showed significantly less Gbp1 recruitment to the *T. gondii* PVM than wild-type Irgm2-reconstituted cells (Fig 6D and E). When localization of the Irgm2 KA mutant was tested, the Irgm2 KA mutant was detected on the *T. gondii* PVM in a manner similar to wild-type Irgm2 (Fig 6D and G), which suggests that ubiquitination of Irgm2 is not involved in localization of itself on parasites. Next, we compared ubiquitination accumulation in Irgm2-deficient cells reconstituted with wild-type Irgm2 and the KA mutant (Fig 6H). Interestingly, ubiquitin accumulation in Irgm2 KA mutant-reconstituted cells was rapidly decreased at later time points in comparison with wild-type Irgm2 (Fig 6H). However, it remained unclear whether Irgm2 on PVM is ubiquitinated. Because Irgm1/Irgm3 DKO cells or Irgb6-deficient cells are severely defective for IFN-γ–induced PVM ubiquitination (Lee et al, 2015, 2020), we compared recruitment of the Irgm2 KA mutant and ubiquitination on the PVM in cells that lacked Irgm1/Irgm3 or Irgb6 (Fig S5A–E). When wild-type Irgm2 and the KA mutant were ectopically expressed in Irgm1/Irgm2/Irgm3 TKO cells or Irgb6-deficient cells (Fig S5A), the Irgm2 KA mutant and wild-type Irgm2 were comparably detected on the PVM in either cell type (Fig S5B and C). In sharp contrast, IFN-γ–induced ubiquitin loading was not detected in Irgm2 KA mutant-reconstituted Irgm1/Irgm2/Irgm3 TKO cells or Irgb6-deficient cells (Fig S5D and E). Thus, the Irgm2 KA mutant normally localized on the PVM in cells defective for IFN-γ–induced PVM ubiquitination, which suggests that Irgm2 on PVM may not be ubiquitinated. When interactions between Gbp1, Irgb6, and Irgm2 were examined, we found that wild-type or C358A Irgm2 were coprecipitated with both Gbp1 and Irgb6 (Fig 6I). Conversely, the KA Irgm2 mutant only associated with Irgb6 but not with Gbp1 (Fig 6I). The M77A mutant did not interact with Irgb6 or Gbp1, which suggests that the capacity for Irgm2 binding to Gbp1 and Irgb6 correlates with recruitment of these effectors to the *T. gondii* PVM. Wild-type Irgm2 was localized at the Golgi apparatus in uninfected cells (Fig S3) (Zhao et al, 2010). However, the KA mutant as well as M77A or C358A mutants did not localize at the Golgi apparatus (Fig S3), which suggests the important role of ubiquitination on Irgm2 in Golgi localization. Taken together, these results indicate that ubiquitinated Irgm2 in the cytosol interacts with Gbp1 to promote recruitment of Gbp1 to the *T. gondii* PVM.

---

detected; *P < 0.05, **P < 0.01, ***P < 0.001. Difference in *T. gondii* inhibition activity between IFN-γ–activated versus nonactivated was subjected to two-way ANOVA, with Tukey's multiple comparisons test to analyze the difference between genotypes. Effector recruitment comparison between genotypes applied one-way ANOVA (Tukey's multiple comparisons test). White arrows indicate recruitment of effector on *T. gondii* PV. Scale bars on microscope images represent 10 μm.

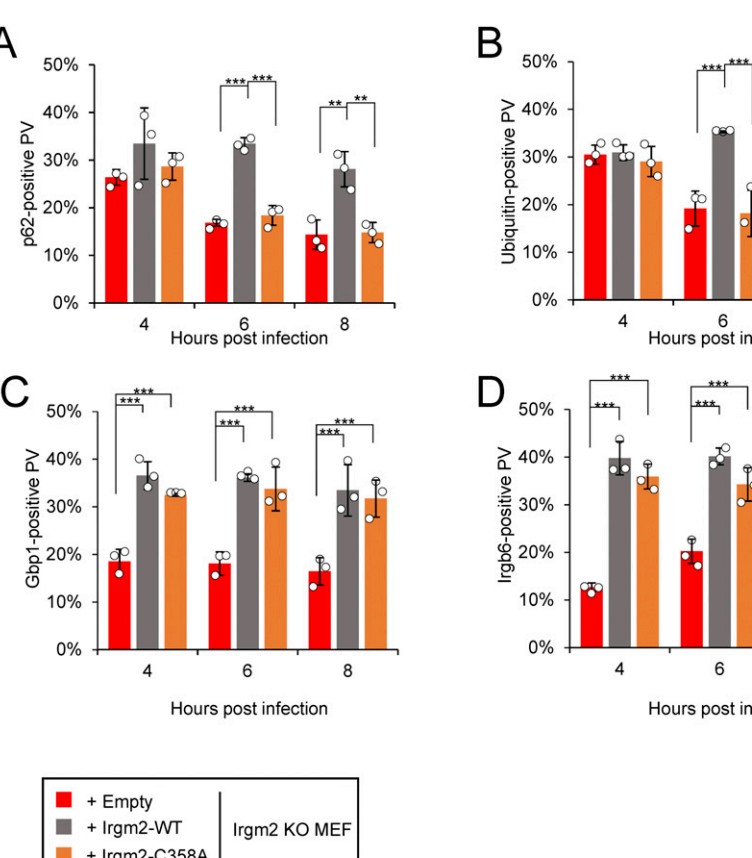

**Figure 5. The Cys358 of Irgm2 is indispensable for prolonged ubiquitin coating and p62 accumulation.**
**(A, B, C, D)** Percentages of recruitment of effectors p62 (A), ubiquitin (B), Gbp1 (C), and Irgb6 (D) on intracellular *Toxoplasma gondii* parasitophorous vacuole in IFN-γ−stimulated Irgm2 KO MEFs reconstituted with the Irgm2 variations (empty vector: red, Irgm2 WT: grey, and Irgm2 C358A: orange) at indicated time points after *T. gondii* infection. All graphs show the mean ± SEM in three independent experiments. All images are representative of three independent experiments. N.D., not detected; *P < 0.05, **P < 0.01, ***P < 0.001. Difference in *T. gondii* inhibition activity between IFN-γ−activated versus nonactivated was subjected to two-way ANOVA, with Tukey's multiple comparisons test to analyze the difference between genotypes. Effector recruitment comparison between genotypes applied one-way ANOVA (Tukey's multiple comparisons test) or two-way ANOVA when time and genotype is considered.

### Irgm2-deficient mice are highly susceptible to *T. gondii* infection

We finally examined the role of Irgm2 in the anti−*T. gondii* response in vivo (Fig 7A–C). Irgm2-deficient mice were infected with *T. gondii* that expressed luciferase by which the parasite dissemination could be measured by an in vivo imaging system (Fig 7A). At day 5 post-infection, Irgm2-deficient mice contained much higher luminescence emitted from luciferase-expressing *T. gondii* than wild-type mice (Fig 7A). When the survival rate was assessed, Irgm2-deficient mice displayed high susceptibility to *T. gondii* (Fig 7B). In addition, the mortality of infected Irgm2-deficient mice was similar to that in IFN-γ−deficient mice, of which all died at day 9 (Fig 7B). Furthermore, wild-type mice recovered fully from the infection (Fig 7B). When parasite numbers in tissues of infected animals were measured, Irgm2-deficient mice showed increased parasite loads in all tested tissues compared with wild-type mice (Fig 7C). Moreover, the parasite numbers in tissues of Irgm2-deficient mice were comparable with those in IFN-γ−deficient mice (Fig S6A), which indicates that Irgm2 critically controls host defense against *T. gondii* infection.

## Discussion

In the present study, we demonstrate that regulatory IRG Irgm2 plays a pivotal role in anti−*T. gondii* cell-autonomous immunity. At the molecular level, we characterized three features of Irgm2: the

GMS configuration of the N-terminal GTPase domain, the C-terminal Cys358 for PVM targeting, and ubiquitination on the lysine residues. The M77A mutation severely affected Irgm2-induced anti−*T. gondii* killing, recruitment of Irgb6 and Gbp1, and Irgm2 localization on the *T. gondii* PVM, which indicates that the GMS configuration of the N-terminal GTPase domain plays a central role in Irgm2-mediated anti−*T. gondii* cell-autonomous immunity. The Irgm1 GMS mutant profoundly reduces affinity for GTP and impairs GTPase functions (Taylor et al, 1997; Martens et al, 2004). In addition, the GTPase domain of IRG proteins plays a role in dimerization of IRGs (Haldar et al, 2013). Considering that Irgm2 deficiency only affected Irgb6 recruitment, the GMS configuration of Irgm2 might be important for heterodimerization with Irgb6 through the N-terminal GTPase domain. In contrast to the M77A mutation, the C358A mutation adjacent to the C-terminal αK specifically impaired Irgm2 localization on the *T. gondii* PVM. In the case of Irgm1, a tight cluster of Cys residues adjacent to the αK is palmitoylated and important for its localization on host mitochondria and the Golgi apparatus (Henry et al, 2014). Although we do not have direct evidence of protein modification of Cys358 in Irgm2, it is plausible to postulate that modification on Cys358, including palmitoylation, might determine Irgm2 localization on the *T. gondii* PVM. Although the Irgm2 KA mutant localized at the *T. gondii* PVM, the percentages of parasites coated with ubiquitin at later time points in Irgm2 KA mutant-reconstituted cells was comparable with those in C358A mutant-reconstituted cells but significantly less than those in wild-type

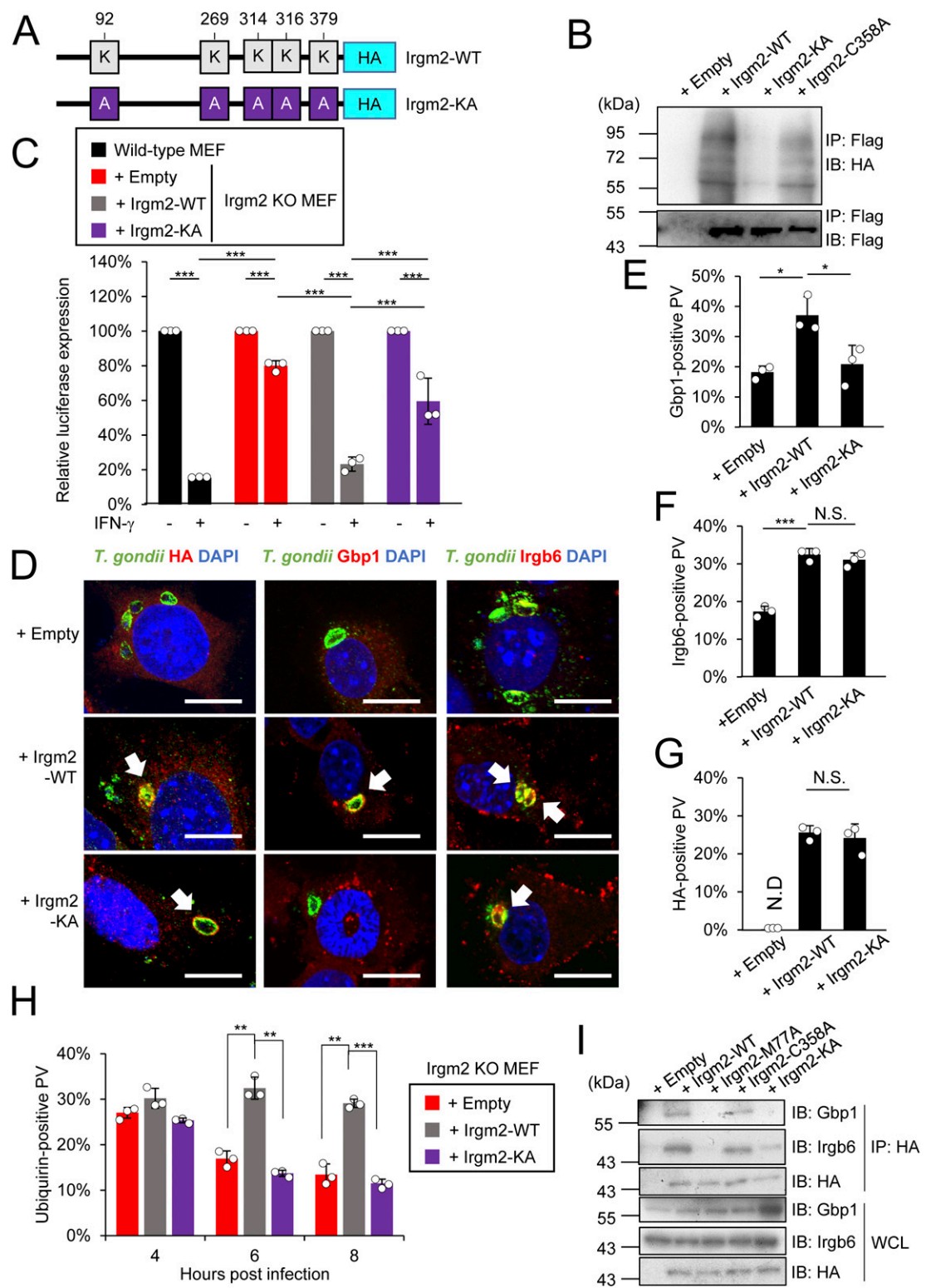

**Figure 6. Irgm2 ubiquitination regulates Gbp1-mediated *Toxoplasma gondii* killing.**
**(A)** Schematic overview of the substitution mutation site on the Irgm2 sequence. **(B)** Western blot image of showing ubiquitin-3xHA immunoprecipitated with Flag tagged Irgm2. Irgm2 KO MEFs stably expressing ubiquitin-3xHA and indicated Irgm2 variant. **(C)** *T. gondii* survival rate in the indicated Irgm2 reconstitution in Irgm2 KO MEFs with IFN-γ stimulation relative to those without IFN-γ treatment by luciferase analysis at 24 h post infection. **(D)** Confocal microscope images to show the localization of Irgm2-HA, Gbp1, and Irgb6 (red) to *T. gondii* parasitophorous vacuole (green), and DAPI (blue) at 4 h post infection in IFN-γ treated Irgm2-KO MEFs reconstituted with indicated Irgm2 variants. **(E, F, G)** Percentages of recruitment of Gbp1 (E), Irgb6 (F), and HA (G). **(H)** *T. gondii* survival rate in the indicated Irgm2 reconstitution in Irgm2 KO

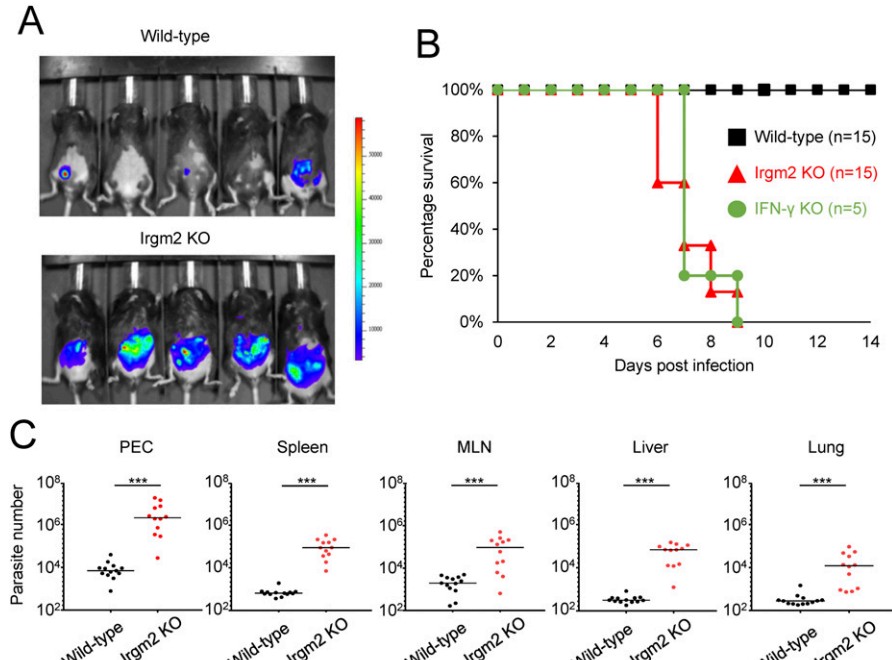

**Figure 7.   Irgm2 KO mice are highly susceptible to _Toxoplasma gondii_ infection.**
**(A)** In vivo bioluminescence imaging comparing between WT and Irgm2 KO mice on day 5 post infection. **(B)** Survival rate of _T. gondii_ infected WT, IFN-γ KO, and Irgm2 KO mice. **(C)** Parasite number in the various tissues collected from infected mice on day 5 post infection. **(B, C)** The data are combined results of three independent experiments (B, C). **(A)** Images are representative of three independent experiments (A). Survival duration comparison between two groups was analyzed by Kaplan–Meier survival analysis log-rank test. Comparison of infection burden between WT versus Irgm2-KO was analyzed with Mann–Whitney Test. *$P < 0.05$, **$P < 0.01$, ***$P < 0.001$.

cells. However, Irgm2 KA mutant-reconstituted Irgm1/Irgm2/Irgm3 TKO cells exhibited recruitment of the KA mutant comparably with wild-type Irgm2 but showed no ubiquitination on the PVM, which suggests that Irgm2 itself is not ubiquitinated at the PVM. At present, the mechanism by which Irgm2 is ubiquitinated remains unclear. Future studies may reveal the molecular mechanism and responsible ubiquitin ligase. Moreover, considering that regulatory IRGs in un-infected cells localize at the Golgi or ER (Martens et al, 2005), Irgm2 might control other membrane trafficking events. We could not dissect the biological significance of Irgm2 localization on the _T. gondii_ PVM in this study. Because Irgm2 localization on the _T. gondii_ PVM was required for prolonged recruitment of p62 and ubiquitin to the vacuole without affecting parasite killing, it would be of interest to examine the role of Irgm2 localization on the _T. gondii_ PVM in parasite killing-independent anti–_T. gondii_ responses in the future.

Irgm2-deficient cells showed specific defects in recruitment of Irgb6 and Gbp1 but not Irga6 or Gbp2, which suggests selective requirement of Irgm2 in Irgb6- and Gbp1-dependent anti–_T. gondii_ responses. Conversely, Irgm1 and Irgm3 globally control the anti–_T. gondii_ cellular response that involves Irga6, Irgb6, Gbp1, and Gbp2 (Lee et al, 2015). Thus, Irgm2 uniquely participates in IFN-inducible GTPase-mediated cell-autonomous immunity against _T. gondii_. Considering that Irgm2 interacts with Irgb6 but not Irga6 (Hunn et al, 2008), a regulatory role of Irgm2 in the effector function of Irgb6 may be probable. However, defective Gbp1 recruitment in Irgm2-deficient cells may be dependent or independent of Irgb6. It has

been previously shown that Irgb6 controls Gbp1 accumulation on the _T. gondii_ PVM (Lee et al, 2020), which suggests a direct link between Irgb6 and Gbp1.

The interaction of ubiquitinated Irgm2 with Gbp1 occurs in the cytosol. We found that Gbp1 loading on the PVM was decreased in ubiquitination-defective Irgm2 KA mutant-reconstituted Irgm2-deficient cells. In addition, the KA Irgm2 mutant did not interact with Gbp1. The Irgm2 C358A mutant associated with Gbp1. In addition, reconstitution of the Irgm2 C358A mutant recovered the loading of Gbp1. The Irgm2 C358A mutant did not localize at the PVM or Golgi apparatus. Collectively, ubiquitinated Irgm2 must interact with Gbp1 to regulate Gbp1 recruitment to the _T. gondii_ PVM in the cytosol but not at the PVM or Golgi apparatus. Irgm2 ubiquitination in the cytosol is required for Gbp1 recruitment to the PVM, its damage, and subsequent ubiquitination. The rapid reduction of ubiquitin accumulation in Irgm2 KA mutant-reconstituted cells may be not due to Irgm2 ubiquitination on the PVM but to an indirect effect by the failure of Gbp1-mediated PVM disruption.

Cells that lack Irgm1/Irgm3 or Atg proteins display severely impaired accumulation of effector IRGs, GBPs, ubiquitin, and p62 (Haldar et al, 2013; Ohshima et al, 2014; Lee et al, 2015), which indicates that regulatory IRGs and Atg proteins play a positive role in localizing effector proteins on the _T. gondii_ PVM in IFN-γ–stimulated cells. In sharp contrast, neither regulatory IRGs nor Atg proteins control Irgm2 localization on the PVM. There is a coordinated loading mechanism of effector IRGs and GBPs on the _T. gondii_ PVM

---

MEFs with IFN-γ stimulation relative to those without IFN-γ treatment by luciferase analysis at 24 h post infection. **(I)** Western blot image of showing Gbp1 and Irgb6 immunoprecipitated with HA tagged Irgm2 of the indicated variants stably expressed in Irgm2 KO MEFs. All graphs show the mean ± SEM in three independent experiments. All images are representative of three independent experiments. N.D., not detected; *$P < 0.05$, **$P < 0.01$, ***$P < 0.001$ IFN-γ–activated versus nonactivated and different antigen used in the co-culture used two-way ANOVA. Comparison between genotypes and different time points applied two-way ANOVA (Tukey's multiple comparisons test). White arrows indicate recruitment of effector on _T. gondii_ parasitophorous vacuole. Scale bars on microscope images represent 10 μm.

after formation (Khaminets et al, 2010; Lee et al, 2020). However, our current study indicates that Irgm2 is out of the temporal hierarchy that governs all other known IFN-γ–inducible effectors for their localization on the *T. gondii* PVM. Considering that localization of reconstituted wild-type Irgm2 on the *T. gondii* PVM in Irgm2-deficient cells requires IFN-γ stimulation, unidentified IFN-γ–inducible host factor(s) and upstream event(s) may be required for Irgm2 recruitment to the PVM.

Irgm2 is essential for anti–*T. gondii* host defense in vivo. Mice that lack Irgm2 were highly susceptible to *T. gondii* infection in a similar manner to IFN-γ–deficient mice, which indicates a nonredundant essential function of Irgm2 in anti–*T. gondii* host defense. Compared with other regulatory IRGs, considering that the time course of death of Irgm2-deficient mice after *T. gondii* infection was similar to that of Irgm3/iNOS double-deficient mice (Zhao et al, 2009), Irgm2-deficient mice might be more susceptible to *T. gondii* than Irgm3-deficient mice in vivo. We found that Irgm2 deficiency led to partially defective translocation of Gbp1 and Irgb6 or reduced retention of p62 and ubiquitin on the *T. gondii* PVM. However, considering that Gbp1-deficient mice have a relatively moderate increase in their in vivo susceptibility to *T. gondii* (Selleck et al, 2013), the partial defects in Gbp1 recruitment to the *T. gondii* PVM by Irgm2 deficiency might not be sufficient to explain the severe in vivo phenotype of Irgm2-deficient mice. Although additional defects in the recruitment of Irgb6, p62, and ubiquitin to the *T. gondii* PVM may account for the severe phenotype of Irgm2-deficient mice, whether the correlative reduction of IFN-inducible effector recruitment to the *T. gondii* PVM could strictly connect with the causation of the in vivo phenotype should be carefully examined in the future. For example, Irgm2 may have pleiotropic functions such as membrane trafficking at the Golgi. Moreover, recent studies have demonstrated a role for Irgm2 in controlling LPS-mediated caspase-11 activation (Eren et al, 2020; Finethy et al, 2020). Although the role of caspase-11 in *T. gondii* infection in vivo remains unknown, dysregulated caspase-11 activation and defective membrane trafficking might affect the high mortality of Irgm2-deficient mice infected with *T. gondii*. In summary, we have demonstrated that Irgm2 plays an important role in Gbp1/Irgb6-mediated parasite killing in IFN-γ–induced cell-autonomous anti–*T. gondii* immunity.

# Materials and Methods

### Cells, mice, and parasites

Primary MEFs were maintained in DMEM (Nacalai Tesque) supplemented with 10% heat-inactivated FBS (Gibco, Life Technologies), 100 U/ml penicillin (Nacalai Tesque), and 100 μg/ml streptomycin (Nacalai Tesque). MEFs that lack Irgm1/Irgm3, Atg proteins, Irgb6, and Gbp1 are described previously (Lee et al, 2015, 2020; Sasai et al, 2017). Bone marrow-derived macrophages were generated by cultivating BM progenitors isolated from BM in complete medium containing 10% L-cell conditioned medium for 6–7 d. The complete medium comprised 10% heat-inactivated FBS in RPMI 1640 medium (Nacalai Tesque). *T. gondii* were parental PruΔHX, luciferase-expressing PruΔHX, and OVA-expressing PruΔHX. They were maintained in Vero cells by passaging every 3 d in RPMI 1640 supplemented with 2% heat-inactivated FBS, 100

U/ml penicillin, and 100 μg/ml streptomycin. All animal experiments were conducted with approval of the Animal Research Committee of Research Institute for Microbial Diseases in Osaka University.

### Generation of Irgm2-deficient mice by CRISPR/Cas9 genome editing

The target gRNA sequence of Irgm2 was 5′-gagaaagattcagctcccacTGG-3′ (TGG; the PAM sequence) in the N-terminus. The insert fragments of Irmg2 gRNA were generated using the primers Irgm2_gRNA_F 5′-TTAATACGACTCACTATAGGgagaaagattcagctcccacGTTTTAGAGCTAGAAATA-GCAAGTTAAAAT-3′ and gRNA_R 5′-AAAAGCACCGACTCGGTGCCACTTTTT-CAAGTTGATAACGGACTAGCCTTATTTTAACTTGCTATTTCTAGCTCT-3′. T7-transcribed Irgm2 gRNA PCR products were purified in gels and used for subsequent generation of gRNA. MEGAshortscript T7 (Life Technologies) was used to generate the gRNA. mRNA that encoded RNA-guided DNA endonuclease Cas9 was generated by in vitro transcription using the mMESSAGE mMA-CHINE T7 ULTRA kit (Life Technologies). The template was amplified by PCR using pEF6-hCas9-Puro and primers T7Cas9_IVT_F and Cas9_R, and then purified in a gel. The synthesized gRNA and Cas9 mRNA were purified using the MEGAclear kit (Life Technologies) and eluted in RNase-free water (Nacalai Tesque). To generate Irgm2-deficient mice, 6-wk-old female C57BL/6 mice were superovulated and mated with C57BL/6 males. Fertilized one-cell-stage embryos were collected from the oviducts and Cas9-encoding mRNA (100 ng/μl) and gRNA (50 ng/μl) were injected into the cytoplasm as described previously (Sasai et al, 2017). The injected live embryos were transferred into the oviducts of pseudopregnant Institute of Cancer Research females at 0.5 d post-coitus. Heterozygous mice were intercrossed to generate Irgm2-deficient mice for use in the in vivo experiments. Irgm2-deficient mice were born at Mendelian ratios and were healthy. Expression of Irgm2 proteins in primary embryonic fibroblasts was analyzed by Western blotting. Two embryos were used to generate two independent Irgm2-deficient MEF lines that similarly showed defects in anti–*T. gondii* responses (data not shown). Irgm1/Irgm2/Irgm3-TKO mice were generated through genome editing by introducing the same Irgm2-targeting gRNA used to generate Irgm2-deficient mice as described above together with Irgm1- or Irgm3-targeting gRNAs into mouse embryos (Lee et al, 2015). Irgm1/Irgm2/Irgm3-TKO mice were born at Mendelian ratios and were healthy. Expression of Irgm1, Irgm2, and Irgm3 proteins in primary embryonic fibroblasts was analyzed by Western blotting (Fig S6B).

### Cloning and recombinant expression

The region of interest of the cDNA corresponding to the wild-type Irgm2 or indicated point mutants or deletion mutants of Irgm2 (GenBank accession no. NM_019440.3) was synthesized from the mRNA of the spleen of C57BL6 mice using primers Irgm2_F 5′-ga-attcaccATGGAAGAGGCAGTTGAGTCACCTGAG-3′ and Irgm2_R 5′-ctcg-agAGGATGAGGAATGGAGAGTCTCAG-3′. Irgm2 GMS, C357A, and C358A mutants were generated using primers GMS_F 5′-CTGGGGACTCTG-GCAATGGCgcgTCATCTTTCATCAATGCCCT-3′ and GMS_R 5′-AGGGCATT-GATGAAAGATGAcgcGCCATTGCCAGAGTCCCCAG-3′; C357A_F 5′-TAGGT-TTTGACTACATGAAGgcgTGCTTTACCTCTCATCACAG-3′ and C357A_R 5′-CTGT-GATGAGAGGTAAAGCAcgcCTTCATGTAGTCAAAACCTA-3′; C358A_F 5′-GTTTT-GACTACATGAAGTGCgccTTTACCTCTCATCACAGTCG-3′ and C358A_R

5′-CGACTGTGATGAGAGGTAAAggcGCACTTCATGTAGTCAAAAC-3′. KA mutants of Irgm2 were artificially synthesized (FASMAC). PCR products were ligated into the EcoRI/XhoI site of the retroviral pMRX-HA expression vector for retroviral infection. The sequences of all constructs were confirmed by DNA sequencing.

### Mice survival and in vivo parasite imaging

Mice were intraperitoneally infected with PruΔHX *T. gondii* tachyzoites that expressed luciferase (1 × 10⁴ in 200 µl PBS per mouse). Mice survival was monitored for up to 15 d postinfection. For in vivo imaging of parasites, mice were intraperitoneally injected with 3 mg D-luciferin in 200 µl PBS (Promega) on day 5 postinfection. Mice were subjected to inhalation anesthesia by isoflurane (Sumitomo Dainippon Pharma). Abdominal photon emission was assessed during 60 s of exposure by an in vivo imaging system (IVIS Spectrum; Xenogen), followed by analysis with Living Image software (Xenogen).

### Reagents

Antibodies against Irgb6 (T-cell specific GTPase [TGTP]; sc-11079), Irgm1 (LRG-47; sc-11075), Irgm2 (GTPI; sc-11088), and Irgm3 (inducibly expressed GTPase [IGTP]; sc-136317) were purchased from Santa Cruz Biotechnology, Inc. Antibodies against FLAG M2 (F3165), β-actin (A1978), and HA were obtained from Sigma-Aldrich. An Anti-HA 1.1 mouse monoclonal antibody was purchased from BioLegend. Rabbit polyclonal anti-GBP2 and mouse monoclonal anti-p62 (PM045) antibodies were obtained from Proteintech and MBL International, respectively. An anti-ubiquitin mouse monoclonal antibody (FK2; MFK-004) was obtained from Nippon Biotest Laboratories. A mouse monoclonal anti-Irga6 (10D7) antibody was provided by Dr. JC Howard (Instituto Gulbenkian de Ciencia). A rabbit polyclonal anti-GBP1 antibody was provided by Dr JC Boothroyd (Stanford University School of Medicine). A mouse monoclonal anti-GRA2 antibody were provided by Dr. D Soldati-Favre (University of Geneva). A custom anti-Irgm2 antibody was purchased from Cosmobio for microscopy analysis. Recombinant mouse IFN-γ was purchased from PeproTech.

### Western blotting

MEFs and macrophages were stimulated with IFN-γ (10 ng/ml) overnight. The cells were washed with PBS and then lysed with 1× (TNE) Tris/NP-40/EDTA buffer (20 mM Tris–HCl, 150 mM NaCl, 1 mM EDTA, and 1% NP-40) or Onyx buffer (20 mM Tris–HCl, 135 mM NaCl, 1% Triton-X, and 10% glycerol) for immunoprecipitation, which contained a protease inhibitor cocktail (Nacalai Tesque) and sonicated for 30 s. The supernatant was collected, incubated with the relevant antibodies overnight, and then pulled down with Protein G Sepharose (GE) for immunoprecipitation. Samples and/or total protein was loaded and separated in 10% or 15% SDS–PAGE gels. After the appropriate length was reached, the proteins in the gel were transferred to a polyvinyl difluoride membrane. The membranes were blocked with 5% dry skim milk (BD Difco Skim milk) in PBS/Tween 20 (0.2%) at room temperature. The membranes were probed overnight at 4°C with the indicated primary antibodies. After washing with PBS/Tween, the membranes were probed with HRP-conjugated secondary antibodies for 1 h at room temperature and then visualized by Luminata Forte Western HRP substrate (Millipore).

### Measurement of *T. gondii* numbers by a luciferase assay

To measure the number of *T. gondii*, cells were untreated or treated with IFN-γ (10 ng/ml) for 24 h. After the stimulation, the cells were infected with luciferase-expressing PruΔHX *T. gondii* (MOI of 0.5) for 24 h. The infected cells were collected and lysed with 100 µl of 1× passive lysis buffer (Promega). The samples were sonicated for 30 s before centrifugation and 5 µl of the supernatants were collected for luciferase expression reading by the dual-luciferase reporter assay system (Promega) using a GLOMAX 20/20 luminometer (Promega). The in vitro data are presented as the percentage of *T. gondii* survival in IFN-γ–stimulated cells relative to unstimulated cells (control).

To measure the number of *T. gondii* in the peritoneal cavity, mesenteric lymph nodes, spleen, liver, and lungs, these organs were removed on day 5 post-infection. The samples were homogenized and lysed in 1 ml of 1× passive lysis buffer, followed by sonication for 30 s. After centrifugation, luciferase activity was measured using 5 µl of the supernatants collected for luciferase expression reading as described for the in vitro model. The in vivo data are presented as absolute values.

### Immunofluorescence microscopy

MEFs were infected with *T. gondii* (MOI 5 or 2) after stimulation with IFN-γ (10 ng/ml) for 24 h. The cells were infected for the indicated time in the respective figures and then fixed for 10 min in PBS containing 3.7% formaldehyde. Cells were then permeabilized with PBS containing 0.002% digitonin (Nacalai Tesque) and blocked with 8% FBS in PBS for 1 h at room temperature. Next, the cells were incubated with antibodies relevant to the experiments for 1 h at 37°C. After gently washing the samples in PBS, the samples were incubated with Alexa 488– and 594–conjugated secondary antibodies as well as DAPI for 1 h at 37°C in the dark. The samples were then mounted onto glass slides with PermaFluor (Thermo Fisher Scientific) and observed under a confocal laser microscope (FV1200 IX-83; Olympus). Images are shown at ×1,000 magnification (scale bar at 5 or 10 µm as indicated). To measure recruitment rates, 100 vacuoles were observed and the numbers of vacuoles coated with effectors were calculated. The counting was repeated three times (three technical replicates). The mean of the three technical replicates was calculated and shown in each circle. After the independent experiments were repeated three times (three biological replicates), three means (three circles) are shown in each figure.

### Mass spectrometric analysis

MEFs that expressed Spot-tagged Irgm2 were lysed in radio-immunoprecipitation assay (RIPA) buffer (20 mM Hepes-NaOH, pH 7.5, 1 mM EGTA, 1 mM MgCl₂, 150 mM NaCl, 0.25% sodium deoxycholate, 0.05% SDS, and 1% NP-40) containing Complete protease and PhosSTOP phosphatase inhibitors (Roche). The lysates were

incubated with anti-Spot nanobody-coupled magnetic agarose beads (Spot-trap_MA; ChromoTek) at 4°C for 3 h. The beads were washed four times with RIPA buffer and then twice with 50 mM ammonium bicarbonate. Proteins on the beads were digested with 200 ng trypsin (MS grade; Thermo Fisher Scientific) at 37°C for 16 h. The digests were reduced, alkylated, acidified, and desalted with GL-Tip SDB (GL Sciences). The eluates were evaporated and dissolved in 3% acetonitrile (ACN) and 0.1% trifluoroacetic acid. LC–MS/MS analysis of the resultant peptides was performed on an EASY-nLC 1200 UHPLC connected to a Q Exactive Plus mass spectrometer equipped with a nanoelectrospray ion source (Thermo Fisher Scientific). The peptides were separated on a 75 $\mu$m inner diameter × 150 mm C18 reversed-phase column (Nikkyo Technos) with a linear 4–32% ACN gradient for 0–100 min, followed by an increase to 80% ACN for 10 min. The mass spectrometer was operated in data-dependent acquisition mode with the top 10 MS/MS method. MS1 spectra were measured with a resolution of 70,000, an automatic gain control target of $1 \times 10^6$, and a mass range from 350 to 1,500 $m/z$. HCD MS/MS spectra were acquired at a resolution of 17,500, an automatic gain control target of $5 \times 10^4$, an isolation window of 2.0 $m/z$, a maximum injection time of 60 ms, and a normalized collision energy of 27. Dynamic exclusion was set to 10 s. Raw data were directly analyzed against the NCBI nonredundant database restricted to *Mus musculus* using Proteome Discoverer v2.3 (Thermo Fisher Scientific) with the Sequest HT search engine. The search parameters were as follows: (a) trypsin as an enzyme with up to two missed cleavages; (b) precursor mass tolerance of 10 ppm; (c) fragment mass tolerance of 0.02 D; (d) carbamidomethylation of cysteine as a fixed modification; (e) acetylation of the protein N-terminus, oxidation of methionine, di-glycine modification of lysine and phosphorylation of serine, threonine, and tyrosine as variable modifications. Peptides were filtered at a false discovery rate of 1% using the percolator node.

### Statistical analysis

Three points in all graphs represent three means derived from three independent experiments (three biological replicates). All statistical analyses were performed using Prism 7 (GraphPad). In infection assays, differences in the *T. gondii* inhibition activity between IFN-γ activated versus nonactivated were subjected to two-way ANOVA with Tukey's multiple comparisons test to analyze the difference between genotypes. When comparing effector recruitment percentages between different genotypes, ordinary one-way ANOVA was used when there were more than two groups. If there were only two groups, the Mann–Whitney test was applied instead. In cases where the effector recruitment percentage was compared with consideration that it was affected by more than one variable, for example genotype and different time points, the adopted analysis was two-way ANOVA (Tukey's multiple comparisons test). Statistical significance of the difference in the survival of mice between two groups was assessed by Kaplan–Meier survival analysis and the log-rank test.

## Supplementary Information

## Acknowledgements

We thank M Enomoto (Osaka University) for secretarial assistance. This study was supported by the Research Program on Emerging and Re-emerging Infectious Diseases (JP20fk0108137), Japanese Initiative for Progress of Research on Infectious Diseases for Global Epidemic (JP20wm0325010) and the Strategic International Collaborative Research Program (JP20jm0210067) from the Agency for Medical Research and Development (AMED), a Grant-in-Aid for Transformative Research Area (B) (Establishment of PLAMP as a new concept to determine self and nonself for obligatory intracellular pathogens; 20B304), for Scientific Research on Innovation Areas (production, function and structure of neo-self; 19H04809), for Scientific Research (B) (18KK0226 and 18H02642) and for Scientific Research (A) (19H00970) from the Ministry of Education, Culture, Sports, Science and Technology, Fusion Oriented Research for Disruptive Science and Technology (JPMJFR206D) and Moonshot research & development (JPMJMS2025) from Japan Science and Technology Agency, program from Joint Usage and Joint Research Programs of the Institute of Advanced Medical Sciences, Tokushima University, Takeda Science Foundation, Mochida Memorial Foundation, Astellas Foundation for Research on Metabolic Disorders, and Research Foundation for Microbial Diseases of Osaka University.

### Author Contributions

A Pradipta: conceptualization, data curation, formal analysis, validation, investigation, methodology, and writing—original draft, review, and editing.
M Sasai: data curation, formal analysis, and investigation.
K Motani: data curation, investigation, and methodology.
JS Ma: formal analysis, investigation, and methodology.
Y Lee: investigation and methodology.
H Kosako: formal analysis, investigation, and methodology.
M Yamamoto: conceptualization, resources, data curation, formal analysis, supervision, funding acquisition, investigation, project administration, and writing—original draft, review, and editing.

### Conflict of Interest Statement

The authors declare that they have no conflict of interest.

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
