## [Reviewer comments · Life Science Alliance]

Life Science Alliance

Cell-autonomous Toxoplasma killing program requires Irgm2 but not its microbe vacuolar localization

Ariel Pradipta, Miwa Sasai, Kou Motani, Ji Ma, Youngae Lee, Hidetaka Kosako, and Masahiro Yamamoto

DOI: <https://doi.org/10.26508/lsa.202000960>

Corresponding author(s): Masahiro Yamamoto, Research Institute for Microbial Diseases

Review Timeline:

Submission Date:	2020-11-16
Editorial Decision:	2020-11-17
Revision Received:	2021-01-30
Editorial Decision:	2021-03-27
Revision Received:	2021-04-16
Editorial Decision:	2021-05-10
Revision Received:	2021-05-17
Accepted:	2021-05-20

Scientific Editor: Shachi Bhatt

Transaction Report:

Please note that the manuscript was previously reviewed at another journal and the reports were taken into account in the decision-making process at Life Science Alliance.

Referee #1 Review

Report for Author:

In murine cells the cytokine interferon gamma upregulates IRGs and GBPs, which play an important role in the destruction of the parasitophorous vacuole. Therefore, understanding the exact mechanism by which these IRGs and GBPs are recruited to the PVM is important.

In this manuscript "Irgm2 at Toxoplasma-forming vacuole bifurcates the microbe killing and antigen presentation programs" Pradipta et al. show that:

Irgm2 deficiency leads to impaired *T. gondii* killing activity in vitro with selectively decreased recruitment of Irgb6 and Gbp1 on the PVM. Irgm2^{-/-} mice are very susceptible to Toxoplasma demonstrating its in vivo importance. The GMS configuration of Irgm2 is essential for the cell-

autonomous immune function of Irgm2. IRGs, Atg, GBP1, Irgb6 proteins are not involved in Irgm2 localization on *T. gondii* PVM. Cys358 of Irgm2 is a determinant for the localization on the *T. gondii* PVM and for its localization at the Golgi apparatus in uninfected cells. However, neither PVM localization nor Golgi localization is needed for its function as Toxoplasma growth inhibition is restored by this mutant as is GBP1/Irgb6 PVM localization. However, prolonged (8hrs but not 4hrs) accumulation of p62 and ubiquitin is dependent on Irgm2 vacuole localization and presentation of vacuolar antigen to activate CD8 + T cells. Irgm2 is ubiquitinated and ubiquitinated Irgm2 interacts with Gbp1 to promote recruitment of Gbp1 to *T. gondii* PVM.

Although experiments were well performed and analyzed there is not that much novelty compared to what has been previously published.

The authors already published that p62 affects antigen presentation (Lee et al. 2015) and therefore that the reduced p62 on the PVM in the Irgm2^{-/-} cells affect antigen presentation was to be expected. The exact role of Irgm2 was also still not clear to this reviewer as some of the conclusions drawn from the results are inconsistent with previous results from this group. For example, the authors show that the PVM localization of Irgm2 is not important for Toxoplasma growth inhibition or GBP1/Irgb6 PVM localization but at the same time they claim that Irgm2 ubiquitination is important for GBP1 PVM localization and for PVM ubiquitination suggesting that it is mainly Irgm2 which is ubiquitinated at the PVM. It is unclear how these two results are compatible. How can they propose that Irgm2 ubiquitination on the PVM is important while at the same time claiming Irgm2 does not need to be at the PVM for its function? Also, in a previous paper these authors showed that in Irgb6^{-/-} cells there is no vacuole ubiquitination and no p62 recruitment. It is unclear how this can be consistent with the recruitment of Irgb6 not being affected when Irgm2 is not at the PVM but p62 is being affected. Furthermore, these authors show that in Irgb6^{-/-} cells the Irgm2 on PVM is not affected (Fig 3b). However, in previous paper Irgb6^{-/-} has no PVM ubiquitination. Does this mean that Irgb6 somehow mediates Irgm2 ubiquitination? The authors need to do a much better job at integrating all these data into a consistent model.

Other major comments:

Because often the means of 3 different groups are compared with each other the two-tailed t-test does not seem the most appropriate for many of the presented analyses.

Referee #2 Review

Report for Author:

This is a well written and interesting report that addresses a key question about the function of Irgm2 in controlling parasite clearance and in facilitating antigen presentation following infection with *Toxoplasma gondii*. The data are convincing and well-presented and I only have a few suggestions for improvement.

1) In the abstract, and other sections, summary statements about the roles of Irgm2 are somewhat confusing. The data show that Irgm2: 1) is important for control of parasite killing through recruitment of Gbp1 (and Irgb6), 2) this does not rely on Irgm2 being present on the vacuole, 3) Irgm2 needs to be ubiquitinated - which happens in the cytosol (not Golgi or PVM). This complexity makes it difficult to succinctly summarize the role of Irgm2. I suggest the authors might consider a different summary in the abstract (in place of Moreover...). Although, ubiquitination of Irgm2 is important for parasite killing through recruitment of Gbp1 to the PVM, it does not require the presence of Irgm2 at the PVM, suggesting they interact in the cytosol.

2) In the Discussion it would also be good to have a clear statement of how the authors interpret the findings - they seem hesitant to state that the interaction of Irgm2 with Gbp1 must happen in the cytosol, although the mutants that do not localize to the Golgi or PVM support this conclusion. I would suggest they make a stronger conclusion for how these proteins interact and then list the supporting data. As the text currently reads, the reader has to decipher what the authors mean from rather indirect comments.

3) This statement, which appears on page 6, does not reflect the data: "We confirmed that the defects in Irgm2-deficient cells contained effector IRGs and GBPs at levels similar to wild-type cells (Fig. 1D)." Rather, the data show that Gbp1 and Irgb6 are lower while Gbp2 and Irga6 are normal in Irgm2^{-/-} cells.

Referee #3 Review

Report for Author:

Immunity related GTPases (IRG)s are critical mediators of innate host defense to *Toxoplasma* infections in mice and have been extensively studied in this context. The mouse IRG family consists of approximately 20 genes of which a subset encodes for the regulatory IRGM proteins *Irgm1*, *Irgm2* and *Irgm3*. Whereas previous work was focused on the role of *Irgm1* and *Irgm3*, the current MS by Ariel et al. studies the role of *Irgm2* in host defense to *Toxo* infections. The MS demonstrates that *Irgm2* - similar to its paralogs *Irgm1/Irgm3* - provides resistance to *Toxoplasma* infections in IFN γ primed mouse cells. The MS further shows that *Irgm2* KO mice - similar to what has been reported for *Irgm1/m3* KO mice - are highly susceptible to *Toxoplasma* infections in vivo. The paper confirms previous observations that *Irgm2* localizes to the *Toxoplasma* parasitophorous vacuoles (PVs). In analogy to mutants generated for other *Irgm* paralogs, the authors demonstrate that an *Irgm2* mutant predicted to be defective for membrane binding, fails to associate with membrane compartments inside the cells, i.e. the Golgi or PVs, as predicted. Yet, the same mutant is nonetheless able to promote the translocation of effector IRGs to PVs and to promote host resistance. A mutant in the GMS motif of the *Irgm2* G domain on the other hand is defective for translocation of effector IRGs to PVs. These observations are consistent with a model originally proposed by Hunn et al. (2008) according to which individual IRGM proteins control the function of specific effector IRGs through direct interactions of the respective G domains of regulatory IRGM and effector IRGs. While these studies up to this point add value to the literature, they only provide an incremental advance in knowledge.

The more provocative idea proposed in this MS is the concept that *Irgm2* at the PVs promotes antigen presentation via PV ubiquitination. Unfortunately, the data presented here do not convincingly support this model, as these conclusions are entirely based on the data shown in Fig 5E, in which a membrane-binding- deficient *Irgm2* mutant fails to complement *Irgm2* KO cells for what amounts to a very minor boost in T cell activation. Here, the study lacks depth, as it fails to establish that *Irgm2* cells and mice indeed have a notable antigen presentation defect. Moreover, an *Irgm2* mutant lacking membrane binding is expected to have pleiotropic effects (keep in mind that the mutant also fails to bind to the Golgi) and deficiencies of the membrane-binding- deficient *Irgm2* mutant are not necessarily the result of a failure of *Irgm2* to associate with PVs. Moreover, *Irgm2* KO cells have a very minor PV ubiquitination phenotype, which is likely due to the absence of ubiquitinated *Irgm2* accumulating at the PV in WT cells. Here, the study fails to test whether *Irgm2* ubiquitination is required for antigen presentation. Lastly, the study provides no mechanism by which *Irgm2* could promote antigen presentation.

Major concerns

- although *Irgm2* KO cells convey significant IFN γ -induced resistance to *Toxoplasma* infection in vitro (Figure 1), *Irgm2* KO mice succumb to *Toxoplasma* infections at the same rate as IFN γ KO mice (Figure 7). This is surprising and needs to be explained. Either *Irgm2* KO have a loss of resistance comparable to IFN γ KO mice in vivo (but not in vitro) or *Irgm2* KO mice have a defect in disease tolerance (i.e. more disease relative to pathogen burden). To begin to address this question, the investigators should determine whether or not parasitic burden in *Irgm2* KO mice is comparable to parasitic burden in IFN γ KO mice. Since IFN γ KO mice were included in the survival experiment (Fig 7B), these data are most likely already available and should be included in Fig. 7C. There are two possible outcomes: i) *Irgm2* KO and IFN γ KO have equally high Toxo burden in vivo - such a result would prompt the question as to why *Irgm2* is more important for host resistance in vivo than it is in vitro; or ii) *Irgm2* KO mice have lower burden than IFN γ KO mice - which would mean that *Irgm2* reduces disease independent of burden level. Here, it may be worth considering two recent studies demonstrating elevated inflammation in *Irgm2* KO mice infected with Gram-negative bacteria / LPS due to hyperactivation of caspase-11 activation (Finethy et al. PMID: 33124745 and Eren et al. PMID: 33124769). Could similarly pattern recognition receptors other than caspase-11 be hyperactivated in Toxo-infected *Irgm2* KO mice and drive inflammation-based mortality? Regardless, the cause of the high mortality rate of *Irgm2* KO mice is unclear and additional studies are required to address this critical aspect of the work presented here.
- the paper proposes an interesting model in which *Irgm* proteins control 2 distinct pathways to execute 2 separate functions, namely i) Toxo killing and ii) Toxo antigen presentation. The MS further propose that these 2 processes can be genetically uncoupled through the use of the C358A mutant. This key concept of the MS is highlighted in the title of the MS. Data in Fig. 4 nicely demonstrates that the C358A mutant fails to localize with the Toxo PV but still provides resistance. However, these second part of this concept, namely that *Irgm2* through binding to the PV and sustained PV ubiquitination promotes antigen presentation hinges entirely on the data presented in Fig. 5E. There are several concerns with these data/ data interpretation/ scope of studies: 1) the IFN γ concentrations are exceedingly low (25 ng/ ml) , i.e. barely above the level of cytokines produced by T cells co-cultured with MEF / Toxo parental controls - compare this to Fig. 4H in Lee et al (2015) where the same assay was run by the Yamamoto lab and IFN γ concentrations were more than 1 log higher; 2) the panel lacks controls: WT cell controls and cells with a complete defect in PV ubiquitination such as *Irgm1/m3* DKO cells 3) MEFs are not professional APCs and not of relevance for Toxo in vivo infections - the study should determine whether *Irgm2* KO DCs have a defect in antigen presentation (as done by the group in Lee et al. in the analysis of p62 KO and *Irgm1/m3* KO DCs) 4) considering that this is the main conclusion of the paper, the authors should also check whether *Irgm2* KO mice have an in vivo defect in antigen presentation by conducting adoptive transfer experiments of OT-I T cells and measuring tetramer + CD8 cells in vivo, as done by the group previously in Lee et al when they analyzed the phenotype of p62 KO mice;
- last but not least, I have a major conceptual concern: the data in Fig.5A-D show that PV ubiquitination at 4 hpi is equal in *Irgm2* KO cells and *Irgm2* complemented cells. At later time points the percentage of ubiquitinated PVs is moderately reduced in *Irgm2* KO cells. In *Irgm1/m3* KO cells on the other hand there's no detectable PV ubiquitination whatsoever (Lee et al.) - how would a little reduction of PV ubiquitination as seen in *Irgm2* KO cells result in a major antigen presentation phenotype? Is there really a causal relationship or couldn't this simply be correlative? The C358A mutation most likely disrupts *Irgm2* membrane binding (and in support of this the authors show that the C358A mutant also fails to associate with the Golgi in Figure S3). Therefore, the C358A

mutation likely disrupt all Irgm2 functions that require membrane binding and would be expected to have pleiotropic effects. Based on the logic applied by the authors, it could also be argued that Irgm2 mediates antigen presentation by localizing the Golgi. Therefore, this study does not provide any compelling evidence that Irgm2 translocation to the PV and/or Irgm2-dependent sustained PV ubiquitination is required for an antigen presentation. It seems equally if not more likely that Irgm2 (and Irgm1/m3) control other membrane trafficking events promoting antigen presentation that are unrelated to PV ubiquitination

- Figure 6 shows Irgm2 itself is being ubiquitinated. A Irgm2 mutant lacking all lysines (Irgm2-KA) still translocates to the Toxo PV but fails to restore complete PV ubiquitination in Irgm2 KO cells. Therefore, the drop in PV ubiquitination seen in Irgm2 KO cells appears to be simply due to the absence of ubiquitinated Irgm2 from the PV. In other words, Irgm2-dependent PV ubiquitination is simply the accumulation of ubiquitinated Irgm2 at the PV. If Irgm2-dependent PV ubiquitination indeed promotes antigen presentation, as the authors claim, then the Irgm2-KA mutant should be defective for antigen presentation. The authors should test this. However, even if the Irgm2-KA mutant were to have an antigen presentation defect, these data would be difficult to interpret considering that the KA mutant also fails to localize to the Golgi (as stated on page 11 - although not actually shown in Fig. S3)

Minor concerns

- Figure 1: the data show that Irgm1/m2/m3 TKO cells fail to reduce Toxo burden upon IFN γ priming, whereas irgm2 KO cells display an intermediate phenotype. Are the authors suggesting that Irgm2 executes a defense mechanism distinct from the one controlled by Irgm1/3? If so, then the question arises whether Irgm1/m3 DKO cells have an intermediate phenotype similar to Irgm2KOs? Since Irgm1/ m3 DKO cells were used later in the MS and therefore appear available, why not include them in Figure 1?

- The paper demonstrates that a M77A mutation renders Irgm2 non-functional. The mutated methionine is part of the unconventional GMS P-loop sequence of IRGM proteins suggesting that the alanine mutations disrupts nucleotide (GDP) binding - see Hunn et al. (2008). Ideally, the M77A mutant should be biochemically characterized. Regardless, the authors should be more careful in their description of the mutant and distinguish between GTPase activity and nucleotide binding

- Although reasonably well written, the paper needs to be edited for language here and there - to provide one example from page 6: "We confirmed that the defects in Irgm2-deficient cells contained effector IRGs and GBPs at levels similar to wild-type cells (Fig. 1D)." Based on the data shown in Fig. 1D, I believe the authors are trying to make the point here that IRG and GBP effector proteins are expressed at comparable levels in WT and Irgm2 KO cells but this information is not conveyed by the sentence.

- the tandem IRG Irgb1-b2 has indeed been shown to act as a decoy for secreted Toxoplasma ROP5/18 kinase complex (Lilue et al., 2013). However, this evolutionary adaptation to overcome microbial immune evasion is not the primary function of these proteins, which are believed to mainly provide host resistance by directly targeting microbes or microbe-containing vacuoles for destruction. Without any discussion of the literature the 'decoy' designation is confusing. I'd suggest to either refer to GKS proteins simply as 'effectors,' or alternatively provide some more background, so that the non-expert reader understands why IRGs are here being referred to as both effectors and decoys. Please, cite Lilue et al., 2013 PMID: 24175088

- The review article on IRGs published by Martens and Howard in 2006 is a bit dated - I'd suggest to also cite a more recent review by Pilla-Moffett et al. (2016) PMID: 27181197

- Please cite Mukhopadhyay et al. (2020) PMID: 32293748 which made observations similar to Haldar et al. (2015) by demonstrating a role for TRAF6 (and TRAF2) in PVM ubiquitination and

TRAF6-dependent killing of Toxo in IFN γ -primed mouse fibroblasts

- The Irgm1/m3-mediated protection of host endomembranes was demonstrated by Haldar et al. (2013) PMID: 23785284 - please cite in introduction when protection of endomembranes is discussed
- Please also cite Park et al. (2016) PMID: 27172324 for the role the ATG system plays in distributing IRGs and GBPs
- Two recently published papers reported 2 novel Irgm2 KO mouse models and described a role for Irgm2 in controlling LPS-mediated caspase-11 activation (Finethy et al. PMID: 33124745 and Eren et al. PMID: 33124769). It seems to make sense for the authors to discuss and contrast their findings with these recent reports.
- There is no description of the TKO cells/ mouse in Materials and Methods - please, provide this information
- Fig. 5A-D lack controls: i.e. WT cells and cells deficient for Ub loading (e.g. Atg3 KO or Irgm1/m3 KOs that were used in a previous publication from the lab by Lee et al. (2015))
- Figure 6C: The authors should provide the p-value of the comparison of empty vector/ + γ vs KA/ + γ

November 17, 2020

Re: Life Science Alliance manuscript #LSA-2020-00960-T

Prof. Masahiro Yamamoto
Research Institute for Microbial Diseases
Osaka University
3-1, Yamadaoka
Suita city, Osaka 565-0871
Japan

Dear Dr. Yamamoto,

Thank you for transferring your manuscript entitled "Irgm2 at Toxoplasma-forming vacuole bifurcates the microbe killing and antigen presentation programs" to Life Science Alliance.

For a brief overview, the manuscript was reviewed at another Alliance journal, where it was rejected as the reviewers were concerned about the conceptual advance. The study was then referred to Life Science Alliance, where the editors found the advance sufficient, and offered the authors further consideration pending following revisions,

- + please address Rev 1 major point 1 and Rev 3 pt 1
- + please tone down the conclusions related to Fig 5 and 6, based on reviewer 3's concerns (pt 2 and 3)
- + please address the minor concerns raised by Rev 2 and Rev 3, and the concern about statistical analysis raised by Rev 1
- + please provide a detailed point-by-point rebuttal to the points raised by the reviewers of the previous journal

Thank you for this interesting contribution to Life Science Alliance. We are looking forward to receiving your revised manuscript.

Sincerely,

Shachi Bhatt, Ph.D.
Executive Editor
Life Science Alliance
<https://www.lsjournal.org/>
Tweet @SciBhatt @LSAJournal

- A letter addressing the reviewers' comments point by point.
- An editable version of the final text (.DOC or .DOCX) is needed for copyediting (no PDFs).
- High-resolution figure, supplementary figure and video files uploaded as individual files: See our detailed guidelines for preparing your production-ready images, <https://www.life-science-alliance.org/authors>
- Summary blurb (enter in submission system): A short text summarizing in a single sentence the study (max. 200 characters including spaces). This text is used in conjunction with the titles of papers, hence should be informative and complementary to the title and running title. It should describe the context and significance of the findings for a general readership; it should be written in the present tense and refer to the work in the third person. Author names should not be mentioned.

B. MANUSCRIPT ORGANIZATION AND FORMATTING:

Point-by-Point responses to Editor:

+ please address Rev 1 major point 1 and Rev 3 pt 1

We modify the sentences, add the new data in **Fig. S5A, S5B, S5C, S5D, S5E and S6A** to address the reviewer #1's major point 1) and the reviewer #3's major point 1). We also remove old Fig. 7D in the revised manuscript to tone down the conclusion related to Figs. 5 and 6.

+ please tone down the conclusions related to Fig 5 and 6, based on reviewer 3's concerns (pt 2 and 3)

In response to the Reviewer #3's major points 2) and 3), we fairly tone down our claim on the role of Irgm2 in antigen presentation and have changed the title of our manuscript as follows;

“Cell-autonomous *Toxoplasma* killing program requires Irgm2 but not its microbe vacuolar localization”

+ please address the minor concerns raised by Rev 2 and Rev 3, and the concern about statistical analysis raised by Rev 1

We add the new data in **Fig. S1C, S3B and S6B** to address the reviewer #1's major point 1) and the reviewer #3's major point 1). In response to minor concerns by Rev2, we amend the **ABSTRACT** and add the **DISCUSSION**.

+ please provide a detailed point-by-point rebuttal to the points raised by the reviewers of the previous journal

The point-by-point responses to reviewers are following.

Point-by-Point responses to Referee #1:

In murine cells the cytokine interferon gamma upregulates IRGs and GBPs, which play an important role in the destruction of the parasitophorous vacuole. Therefore, understanding the exact mechanism by which these IRGs and GBPs are recruited to the PVM is important. In this manuscript " Irgm2 at Toxoplasma-forming vacuole bifurcates the microbe killing and antigen presentation programs "Pradipta et al. show that:

Irgm2 deficiency leads to impaired T. gondii killing activity in vitro with selectively decreased recruitment of Irgb6 and Gbp1 on the PVM. Irgm2-/- mice are very susceptible to Toxoplasma demonstrating its in vivo importance. The GMS configuration of Irgm2 is essential for the cell-autonomous immune function of Irgm2. IRGs, Atg, GBP1, Irgb6 proteins are not involved in Irgm2 localization on T. gondii PVM. Cys358 of Irgm2 is a determinant for the localization on the T. gondii PVM and for its localization at the Golgi apparatus in uninfected cells. However, neither PVM localization nor Golgi localization is needed for its function as Toxoplasma growth inhibition is restored by this mutant as is GBP1/Irgb6 PVM localization. However, prolonged (8hrs but not 4hrs) accumulation of p62 and ubiquitin is dependent on Irgm2 vacuole localization and presentation of vacuolar antigen to activate CD8+ T cells. Irgm2 is ubiquitinated and ubiquitinated Irgm2 interacts with Gbp1 to promote recruitment of Gbp1 to T. gondii PVM.

Although experiments were well performed and analyzed there is not that much novelty compared to what has been previously published.

We thank the reviewer's constructive suggestions and would like the reviewer to assess our revised manuscript as a candidate for Life Science Alliance.

The authors already published that p62 affects antigen presentation (Lee et al. 2015) and therefore that the reduced p62 on the PVM in the Irgm2-/- cells affect antigen presentation was to be expected. The exact role of Irgm2 was also still not clear to this reviewer as some of the conclusions drawn from the results are inconsistent with previous results from this group.

For example, the authors show that the PVM localization of Irgm2 is not important for Toxoplasma growth inhibition or GBP1/Irgb6 PVM localization but at the same time they claim that Irgm2 ubiquitination is important for GBP1 PVM localization and for PVM ubiquitination suggesting that it is mainly Irgm2 which is ubiquitinated at the PVM. It is unclear how these two results are compatible. How can they propose that Irgm2 ubiquitination on the PVM is important while at the same time claiming Irgm2 does not need to be at the PVM for its function? Also, in a previous paper these authors showed that in Irgb6-/- cells there is no vacuole ubiquitination and no p62 recruitment. It is unclear how this can be consistent with the recruitment of Irgb6 not being affected when Irgm2 is not at the PVM but p62 is being affected. Furthermore, these authors show that in Irgb6-/- cells the Irgm2 on PVM is not affected (Fig 3b). However, in previous paper Irgb6-/- has no PVM ubiquitination. Does this mean that Irgb6 somehow mediates Irgm2 ubiquitination? The authors need to do a much better job at integrating all these data into a consistent model.

We are grateful with the reviewer's comprehensive assessment and would like to clarify that Irgm2 ubiquitination does not occur on the PVM due to the new data in this revision (see below). To answer this reviewer's question, we further examined whether Irgm2 on the PVM is ubiquitinated. Since we have shown that Irgm1/Irgm3 DKO cells or Irgb6-deficient cells were severely defective in IFN- γ -induced PVM ubiquitination (Lee et al. Cell Rep. 2015; Lee et al. Life Sci Alliance. 2020), we compared recruitment of the Irgm2 KA mutant and ubiquitination on PVM in cells lacking Irgm1/Irgm3 or Irgb6 (**Fig. S5A-S5E**). When wild-type Irgm2 and the KA mutant were ectopically expressed in Irgm1/Irgm2/Irgm3-TKO cells or Irgb6-deficient cells (**Fig. S5A**), the Irgm2 KA mutant as well as wild-type Irgm2 was comparably detected on the PVM in either cell type (**Fig. S5B and S5C**). On the other hand, ubiquitination on the PVM was not observed or severely impaired in Irgm1/Irgm2/Irgm3-TKO cells or Irgb6-deficient cells expressing the Irgm2 KA mutant, respectively (**Fig. S5D and S5E**). Thus, the Irgm2 KA mutant could normally localize on the PVM in cells defective in IFN- γ -induced PVM ubiquitination, suggesting that Irgm2 on PVM may not be ubiquitinated. Although we previously showed that Irgb6-deficient cells exhibit no PVM ubiquitination, non-ubiquitinated Irgm2 might localize at the PVM in Irgb6-deficient or Irgm1/Irgm3 DKO cells. Also, Irgm2 ubiquitination for the proper function of Gbp1 might occur

in the cytosol. It is obvious that the original illustration (**old Fig. 7D**), in which the wild-type Irgm2 status on the PVM was ubiquitinated, is not correct. In addition, to tone down the specific function of Irgm2 in antigen presentation due to the request by the LSA editor, we decide to delete the illustration in this revised manuscript. We add the new data in **Fig. S5A, S5B, S5C, S5D and S5E** with sentences in the **RESULTS** and **DISCUSSION** as bellow;

“**However, it remained clear whether Irgm2 on PVM is ubiquitinated. Since we have shown that Irgm1/Irgm3 DKO cells or Irgb6-deficient cells were severely defective in IFN- γ -induced PVM ubiquitination (Lee *et al.* 2015; Lee *et al.* 2020), we compared recruitment of the Irgm2 KA mutant and ubiquitination on PVM in cells lacking Irgm1/Irgm3 or Irgb6 (Fig. S5A-S5E). When wild-type Irgm2 and the KA mutant were ectopically expressed in Irgm1/Irgm2/Irgm3 TKO cells or Irgb6-deficient cells (Fig. S5A), the Irgm2 KA mutant as well as wild-type Irgm2 was comparably detected on the PVM in either cell type (Fig. S5B and S5C). In sharp contrast, IFN- γ -induced ubiquitin loading was not detected in the Irgm2 KA mutant-reconstituted Irgm1/Irgm2/Irgm3 TKO cells or Irgb6-deficient cells (Fig. S5D and S5E). Thus, the Irgm2 KA mutant could normally localize on the PVM in cells defective in IFN- γ -induced PVM ubiquitination, suggesting that Irgm2 on PVM may not be ubiquitinated.**” in the **RESULTS**, and “**On the other hand, the Irgm2 KA mutant-reconstituted Irgm1/Irgm2/Irgm3 TKO cells exhibited recruitment of the KA mutant as comparable as wild-type Irgm2 but showed no ubiquitination on PVM, suggesting that Irgm2 itself is not an ubiquitin substrate at the PVM.**” in the **DISCUSSION**.

Other major comments:

Because often the means of 3 different groups are compared with each other the two-tailed t-test does not seem the most appropriate for many of the presented analyses.

We thank the reviewer’s suggestion. We have adjusted the statistical tests accordingly. The statistical analysis method is described in the **MATERIALS AND METHODS** as follows;

Statistical analysis

“Three points in all graphs represent three means derived from three independent experiments (three biological replicates). All statistical analyses were performed using Prism 9 (GraphPad). In infection assay, difference in *T. gondii* inhibition activity between IFN- γ activated vs non-activated was subjected to two-way ANOVA, with Tukey’s multiple comparisons test to analyze the difference between genotypes. When comparing effector recruitment percentage between different genotypes, ordinary one-way ANOVA is used if there are more than 2 groups. If there were only 2 groups, Mann-Whitney Test was applied instead. In cases where effector recruitment percentage is compared with consideration that it is affected by more than one variable, for example genotype and different time points, analysis adopted two-way ANOVA (Tukey’s multiple comparisons test). The statistical significance of difference in the survival of mice between 2 groups was analyzed by Kaplan-Meier survival analysis Log rank test.” in the **MATERIALS AND METHODS.**

Point-by-Point responses to Referee #2:

This is a well written and interesting report that addresses a key question about the function of Irgm2 in controlling parasite clearance and in facilitating antigen presentation following infection with Toxoplasma gondii. The data are convincing and well-presented and I only have a few suggestions for improvement.

We thank the reviewer's positive evaluation and would like to respond to his/her suggestions as follows;

1) In the abstract, and other sections, summary statements about the roles of Irgm2 are somewhat confusing. The data show that Irgm2: 1) is important for control of parasite killing through recruitment of Gbp1 (and Irgb6), 2) this does not rely on Irgm2 being present on the vacuole, 3) Irgm2 needs to be ubiquitinated - which happens in the cytosol (not Golgi or PVM). This complexity makes it difficult to succinctly summarize the role of Irgm2. I suggest the authors might consider a different summary in the abstract (in place of Moreover...). Although, ubiquitination of Irgm2 is important for parasite killing through recruitment of Gbp1 to the PVM, it does not require the presence of Irgm2 at the PVM, suggesting they interact in the cytosol.

We are grateful for the reviewer's discussion. We agree with the reviewer's point of view about needing to summarize the role of Irgm2 ubiquitination and Irgm2 localization at the PVM in a clearer fashion. In the revised manuscript, we adjust our abstract to make a better summary in accordance with the reviewer's suggestions as follows;

“Here we show that Irgm2 is important for control of parasite killing through recruitment of Gbp1 and Irgb6, which does not require Irgm2 localization at Toxoplasma PVM. Ubiquitination of Irgm2 in the cytosol but not at the PVM is also important for parasite killing through recruitment of Gbp1 to the PVM. In contrast, the PVM ubiquitination and p62/Sqstm1 loading at later time points post Toxoplasma infection

requires the Irgm2 localization at the PVM. Irgm2-deficient mice are highly susceptible to Toxoplasma infection. Taken together, these data indicate that IFN-inducible GTPase-dependent cell-autonomous immunity is controlled in a manner dependent or independent on the Irgm2 localization at the Toxoplasma PVM.” in the ABSTRACT.

2) In the Discussion it would also be good to have a clear statement of how the authors interpret the findings - they seem hesitant to state that the interaction of Irgm2 with Gbp1 must happen in the cytosol, although the mutants that do not localize to the Golgi or PVM support this conclusion. I would suggest they make a stronger conclusion for how these proteins interact and then list the supporting data. As the text currently reads, the reader has to decipher what the authors mean from rather indirect comments.

We appreciate the reviewer’s insight and apologize for our failure to emphasize the point. In the revised manuscript, we first include a suggestive statement that interaction between Irgm2 and Gbp1 happens in the cytosol and follow this statement by supporting data as follows;

“The interaction of the ubiquitinated Irgm2 with Gbp1 must happen in the cytosol. We found that Gbp1 loading on the PVM was decreased in the ubiquitination-defective Irgm2 KA mutant-reconstituted Irgm2-deficient cells. In addition, the KA Irgm2 mutant did not interact with Gbp1. The Irgm2 C358A mutant associated with Gbp1. In addition, reconstitution of the Irgm2 C358A mutant recover the loading of Gbp1. The Irgm2 C358A mutant could not localize at the PVM and Golgi apparatus. Collectively, the ubiquitinated Irgm2 must interact with Gbp1 to regulate the Gbp1 recruitment to T. gondii PVM in the cytosol but not at the PVM or Golgi apparatus.” in the DISCUSSION.

3) This statement, which appears on page 6, does not reflect the data: "We confirmed that the defects in Irgm2-deficient cells contained effector IRGs and GBPs at levels similar to wild-type cells (Fig. 1D)." Rather, the data show that Gbp1 and Irgb6 are lower while Gbp2 and Irga6 are normal in Irgm2-/- cells.

We thank the reviewer's notice and apologize for the lack of clarity due to our unclear English writing. Based upon **Fig 1D**, we just tried to report that "the protein expression levels" are comparable at time points tested between WT and Irgm2-KO MEFs. Therefore, we amend the sentence as follows;

"We confirmed that IRG and GBP effector proteins are expressed at comparable levels in wild-type and Irgm2-deficient cells (Fig. 1D)" in the RESULTS.

Point-by-Point responses to Referee #3:

Immunity related GTPases (IRG)s are critical mediators of innate host defense to Toxoplasma infections in mice and have been extensively studied in this context. The mouse IRG family consists of approximately 20 genes of which a subset encodes for the regulatory IRGM proteins Irgm1, Irgm2 and Irgm3. Whereas previous work was focused on the role of Irgm1 and Irgm3, the current MS by Ariel et al. studies the role of Irgm2 in host defense to Toxo infections. The MS demonstrates that Irgm2 - similar to its paralogs Irgm1/Irgm3 - provides resistance to Toxoplasma infections in IFN γ primed mouse cells. The MS further shows that Irgm2 KO mice - similar to what has been reported for Irgm1/m3 KO mice - are highly susceptible to Toxoplasma infections in vivo.

The paper confirms previous observations that Irgm2 localizes to the Toxoplasma parasitophorous vacuoles (PVs). In analogy to mutants generated for other Irgm paralogs, the authors demonstrate that an Irgm2 mutant predicted to be defective for membrane binding, fails to associate with membrane compartments inside the cells, i.e. the Golgi or PVs, as predicted. Yet, the same mutant is nonetheless able to promote the translocation of effector IRGs to PVs and to promote host resistance. A mutant in the GMS motif of the Irgm2 G domain on the other hand is defective for translocation of effector IRGs to PVs. These observations are consistent with a model originally proposed by Hunn et al. (2008) according to which individual IRGM proteins control the function of specific effector IRGs through direct interactions of the respective G domains of regulatory IRGM and effector IRGs. While these studies up to this point add value to the literature, they only provide an incremental advance in knowledge.

The more provocative idea proposed in this MS is the concept that Irgm2 at the PVs promotes antigen presentation via PV ubiquitination. Unfortunately, the data presented here do not convincingly support this model, as these conclusions are entirely based on the data shown in Fig 5E, in which a membrane-binding- deficient Irgm2 mutant fails to complement Irgm2 KO cells for what amounts to a very minor boost in T cell activation. Here, the study lacks depth,

as it fails to establish that Irgm2 cells and mice indeed have a notable antigen presentation defect. Moreover, an Irgm2 mutant lacking membrane binding is expected to have pleiotropic effects (keep in mind that the mutant also fails to bind to the Golgi) and deficiencies of the membrane-binding- deficient Irgm2 mutant are not necessarily the result of a failure of Irgm2 to associate with PVs. Moreover, Irgm2 KO cells have a very minor PV ubiquitination phenotype, which is likely due to the absence of ubiquitinated Irgm2 accumulating at the PV in WT cells. Here, the study fails to test whether Irgm2 ubiquitination is required for antigen presentation. Lastly, the study provides no mechanism by which Irgm2 could promote antigen presentation.

First of all, we thank the reviewer's extensive evaluation and critical comments regarding the specific role of Irgm2 in antigen presentation. We would like the reviewer to assess our revised manuscript as a candidate for Life Science Alliance.

We agree with the reviewer's concern that Irgm2 (and Irgm1/Irgm3) control other membrane trafficking events that could lead to pleiotropic defects including antigen presentation in a manner independent on PV ubiquitination. According to the editor's recommendation, we would like to *tone down* the role of Irgm2 in antigen presentation as described in the following point-by-point responses to her/his major comments 2) and 3).

Major concerns

1) - although Irgm2 KO cells convey significant IFNgamma-induced resistance to Toxoplasma infection in vitro (Figure 1), Irgm2 KO mice succumb to Toxoplasma infections at the same rate as IFNgamma KO mice (Figure 7). This is surprising and needs to be explained. Either Irgm2 KO have a loss of resistance comparable to IFNgamma KO mice in vivo (but not in vitro) or Irgm2 KO mice have a defect in disease tolerance (i.e. more disease relative to pathogen burden). To begin to address this question, the investigators should determine whether or not parasitic burden in Irgm2 KO mice is comparable to parasitic burden in IFNgamma KO mice. Since IFNgamma KO mice were included in the survival experiment (Fig 7B), these data are most likely already available and should be included in

Fig. 7C. There are two possible outcomes: i) Irgm2 KO and IFN γ KO have equally high Toxo burden in vivo - such a result would prompt the question as to why Irgm2 is more important for host resistance in vivo than it is in vitro; or ii) Irgm2 KO mice have lower burden than IFN γ KO mice - which would mean that Irgm2 reduces disease independent of burden level. Here, it may be worth considering two recent studies demonstrating elevated inflammation in Irgm2 KO mice infected with Gram-negative bacteria / LPS due to hyperactivation of caspase-11 activation (Finethy et al. PMID: 33124745 and Eren et al. PMID: 33124769). Could similarly pattern recognition receptors other than caspase-11 be hyperactivated in Toxo-infected Irgm2 KO mice and drive inflammation-based mortality? Regardless, the cause of the high mortality rate of Irgm2 KO mice is unclear and additional studies are required to address this critical aspect of the work presented here.

To address this question, we compared *T. gondii* burden in different organs between WT, Irgm2 KO and IFN- γ KO mice (**Fig. S6A**). We observed that Irgm2 and IFN- γ KO mice had comparable burdens. Therefore we consider that, although Irgm2 KO cells showed a partial defect in IFN- γ -mediated *T. gondii* killing compared with Irgm1/m3 DKO or Irgm1/m2/m3 TKO cells (**Fig. S1C**), Irgm2 is more important for host resistance *in vivo* than *in vitro*. Therefore, the reviewer's speculation i) is likely. As the reviewer suggested below, Irgm2 might play a role in membrane trafficking at Golgi which is involved in pleiotropic functions including negative regulation of caspase-11 activation as well as IRG/GBP-mediated parasite killing. Loss of them might lead to a loss of resistance comparable to IFN- γ KO mice *in vivo*. The new data is included in **Fig. S6A** with sentences in **RESULTS and DISCUSSION** as follows;

“Moreover, the parasite numbers in tissues in Irgm2-deficient mice were comparable to those in IFN- γ -deficient mice (**Fig. S6A**)” in the **RESULTS**, and

“Irgm2 may potentially have pleiotropic functions such as membrane trafficking at Golgi. Moreover, recent studies demonstrate a role for Irgm2 in controlling LPS-mediated caspase-11 activation (Eren et al., 2020; Finethy et al., 2020). Although the role of caspase-11 in *T. gondii* infection in vivo remains to be seen, dysregulated caspase-11 activation as well as defective

membrane trafficking might affect the high mortality in Irgm2-deficient mice in *T. gondii* infection.” in **DISCUSSION**.

2) - the paper proposes an interesting model in which Irgm proteins control 2 distinct pathways to execute 2 separate functions, namely i) Toxo killing and ii) Toxo antigen presentation. The MS further propose that these 2 processes can be genetically uncoupled through the use of the C358A mutant. This key concept of the MS is highlighted in the title of the MS. Data in Fig. 4 nicely demonstrates that the C358A mutant fails to localize with the Toxo PV but still provides resistance. However, these second part of this concept, namely that Irgm2 through binding to the PV and sustained PV ubiquitination promotes antigen presentation hinges entirely on the data presented in Fig. 5E. There are several concerns with these data/ data interpretation/ scope of studies: 1) the IFN γ concentrations are exceedingly low (25 ng/ ml) , i.e. barely above the level of cytokines produced by T cells co-cultured with MEF / Toxo parental controls - compare this to Fig. 4H in Lee et al (2015) where the same assay was run by the Yamamoto lab and IFN γ concentrations were more than 1 log higher; 2) the panel lacks controls: WT cell controls and cells with a complete defect in PV ubiquitination such as Irgm1/m3 DKO cells 3) MEFs are not professional APCs and not of relevance for Toxo in vivo infections - the study should determine whether Irgm2 KO DCs have a defect in antigen presentation (as done by the group in Lee et al. in the analysis of p62 KO and Irgm1/m3 KO DCs) 4) considering that this is the main conclusion of the paper, the authors should also check whether Irgm2 KO mice have an in vivo defect in antigen presentation by conducting adoptive transfer experiments of OT-I T cells and measuring tetramer + CD8 cells in vivo, as done by the group previously in Lee et al when they analyzed the phenotype of p62 KO mice;

According to instruction from the editor of Life Science Alliance, we just respond to the reviewer's comments without further additional experiments. In conclusion, we tone down our argument regarding the role of Irgm2 in antigen presentation since we agree with the reviewer's concern on the potential function of Irgm2 in membrane trafficking that could lead to the defect in antigen presentation. Originally, we had planned to perform the additional experiments in the original submission. However, we had not done them since Irgm2 KO cells are defective not only

in antigen presentation (via PVM ubiquitination and p62) but also in parasite killing (PVM disruption via Gbp1 and Irgb6). In our model, PVM ubiquitination via p62 is not related to parasite killing (Lee et al. *Cell Rep.* (2015)). On the other hand, Irgm1/m3 DKO cells, which had defects in both PVM ubiquitination (hence, antigen presentation) and parasite killing, are not the appropriate positive control to assess the role of the C358A Irgm2 mutant, which could reconstitute parasite killing but not antigen presentation in Irgm2 KO cells. As well, since Irgm2 KO DCs and mice are defective in both antigen presentation and parasite killing, we had considered that experiments 3) and 4) are not appropriate to assess the specific loss function of antigen presentation but not parasite killing in Irgm2 KO DCs and mice. To do so under physiological condition, we had tried to generate C358A Irgm2 knock-in mice but failed due to lack of C358A Irgm2 protein expression with unknown reasons (data not shown). Therefore, we could had just presented a tiny set of data in **Fig. 5E** in the original submission. Anyway, we fairly tone down our argument regarding the role of Irgm2 in antigen presentation in the revised manuscript.

3) - last but not least, I have a major conceptual concern: the data in Fig.5A-D show that PV ubiquitination at 4 hpi is equal in Irgm2 KO cells and Irgm2 complemented cells. At later time points the percentage of ubiquitinated PVs is moderately reduced in Irgm2 KO cells. In Irgm1/m3 KO cells on the other hand there's no detectable PV ubiquitination whatsoever (Lee et al.) - how would a little reduction of PV ubiquitination as seen in Irgm2 KO cells result in a major antigen presentation phenotype? Is there really a causal relationship or couldn't this simply be correlative? The C358A mutation most likely disrupts Irgm2 membrane binding (and in support of this the authors show that the C358A mutant also fails to associate with the Golgi in Figure S3). Therefore, the C358A mutation likely disrupt all Irgm2 functions that require membrane binding and would be expected to have pleiotropic effects. Based on the logic applied by the authors, it could also be argued that Irgm2 mediates antigen presentation by localizing the Golgi. Therefore, this study does not provide any compelling evidence that Irgm2 translocation to the PV and/or Irgm2-dependent sustained PV ubiquitination is required for an antigen presentation. It seems equally if not more likely that Irgm2 (and Irgm1/m3) control other membrane trafficking events promoting antigen presentation that are unrelated to PV ubiquitination

We agree with the reviewer's concern on the potential function of Irgm2 in membrane trafficking that could lead to pleiotropic defects including antigen presentation, and therefore we tone down our argument regarding the role of Irgm2 in antigen presentation in the revised manuscript. The possibility is added in the **DISCUSSION** as follows;

“Loss of the retention of p62 and ubiquitin at the PVM may retard activation of CD8⁺ T cells in the C358A-reconstituted cells. However, given that regulatory IRGs in uninfected cells localize at Golgi or ER (Martens *et al.*, 2005), Irgm2 might control other membrane trafficking events promoting antigen presentation that are unrelated to PV ubiquitination.” in the **DISCUSSION**.

4) - Figure 6 shows Irgm2 itself is being ubiquitinated. A Irgm2 mutant lacking all lysines (Irgm2-KA) still translocates to the Toxo PV but fails to restore complete PV ubiquitination in Irgm2 KO cells. Therefore, the drop in PV ubiquitination seen in Irgm2 KO cells appears to be simply due to the absence of ubiquitinated Irgm2 from the PV. In other words, Irgm2-dependent PV ubiquitination is simply the accumulation of ubiquitinated Irgm2 at the PV. If Irgm2-dependent PV ubiquitination indeed promotes antigen presentation, as the authors claim, then the Irgm2-KA mutant should be defective for antigen presentation. The authors should test this. However, even if the Irgm2-KA mutant were to have an antigen presentation defect, these data would be difficult to interpret considering that the KA mutant also fails to localize to the Golgi (as stated on page 11 - although not actually shown in Fig. S3)

We had originally tested whether the Irgm2-KA mutant could reconstitute the antigen presentation together with the C358A mutant (**Supporting Figure 1**). The ectopic expression of the Irgm2-KA mutant in Irgm2 KO cells had failed to recover the antigen presentation (**Supporting Figure 1**), suggesting that Irgm2 ubiquitination is involved in antigen presentation.

[Figure removed by editorial staff per authors' request]

Supporting Figure 1. Not only the C358A Irgm2 mutant but also the KA mutant is defective in antigen presentation in the reconstituted MEFs.

IFN- γ production by OT-I T cells after 72 hr co-culture with Irgm2 KO MEFs reconstituted with the indicated Irgm2 variations infected with γ -irradiated Pru *T. gondii*-expressing p30-OVA luciferase or parental strain for 7 hr, or treated with 0.1 nM OVA257-264 peptide for 3 hr.

We originally showed and interpreted that “Interestingly, ubiquitin accumulation in the KA Irgm2 KA mutant-reconstituted cells was rapidly decreased at later time points in comparison to wild-type Irgm2 (**Fig. 6H**). Given that, although the Irgm2 KA mutant localizes at *T. gondii* PVM, the ubiquitination at later time points is reduced in the Irgm2 KA mutant-reconstituted cells, Irgm2 might be a substrate of ubiquitin on *T. gondii* PVM at later time points”. However, in response to the Reviewer #1’s suggestion, we found that the reconstitution of Irgm2-KA mutant in Irgm1/Irgm2/Irgm3 TKO cells recovers the recruitment to the PVM but not the PVM ubiquitination (**Fig. S5B, S5C, S5D and S5E**), suggesting that our previous model of Irgm2 ubiquitination at the PVM is unlikely. Given that the Irgm2 ubiquitination in the cytosol is required for Gbp1 recruitment to PVM, PVM damage and the subsequent PVM ubiquitination, the rapid reduction of ubiquitin accumulation in the Irgm2 KA mutant-reconstituted cells may be not due to the Irgm2 ubiquitination at the PVM but to indirect effect by the failure of Gbp1-mediated PVM disruption. The new interpretation is included in the revised manuscript as follows;

“**However, it remained clear whether Irgm2 on PVM is ubiquitinated. Since we have shown that Irgm1/Irgm3 DKO cells or Irgb6-deficient cells were severely defective in IFN- γ**

induced PVM ubiquitination (Lee *et al.*, 2015, 2020), we compared recruitment of the Irgm2 KA mutant and ubiquitination on PVM in cells lacking Irgm1/Irgm3 or Irgb6 (Fig. S5A-S5E). When wild-type Irgm2 and the KA mutant were ectopically expressed in Irgm1/Irgm2/Irgm3 TKO cells or Irgb6-deficient cells (Fig. S5A), the Irgm2 KA mutant as well as wild-type Irgm2 was comparably detected on the PVM in either cell type (Fig. S5B and S5C). In sharp contrast, IFN- γ -induced ubiquitin loading was not detected in the Irgm2 KA mutant-reconstituted Irgm1/Irgm2/Irgm3 TKO cells or Irgb6-deficient cells (Fig. S5D and S5E). Thus, the Irgm2 KA mutant could normally localize on the PVM in cells defective in IFN- γ -induced PVM ubiquitination, suggesting that Irgm2 on PVM may not be ubiquitinated.” in the **RESULTS**, and “The Irgm2 ubiquitination in the cytosol is required for Gbp1 recruitment to PVM, its damage and the subsequent ubiquitination. The rapid reduction of ubiquitin accumulation in the Irgm2 KA mutant-reconstituted cells may be not due to the Irgm2 ubiquitination on PVM but to indirect effect by the failure of Gbp1-mediated PVM disruption” in the **DISCUSSION**.

Minor concerns

- Figure 1: the data show that Irgm1/m2/m3 TKO cells fail to reduce Toxo burden upon IFN γ priming, whereas irgm2 KO cells display an intermediate phenotype. Are the authors suggesting that Irgm2 executes a defense mechanism distinct from the one controlled by Irgm1/3? If so, then the question arises whether Irgm1/m3 DKO cells have an intermediate phenotype similar to Irgm2KOs? Since Irgm1/ m3 DKO cells were used later in the MS and therefore appear available, why not include them in Figure 1?

To answer the reviewer’s question, we compared the parasite killing between Irgm2 KO, Irgm1/m3 DKO and Irgm1/m2/m3 TKO cells. We found that Irgm1/m3 DKO cells showed an intermediate defect between Irgm2 KO and Irgm1/m2/m3 TKO cells. The new data is included in Fig. S1C with sentences as follows;

“However, the magnitude of the defect in Irgm2-single deficient cells was not as severe as Irgm1/Irgm3 double KO (DKO) or Irgm1/Irgm2/Irgm3 triple knockout (TKO) cells (Fig. 1A and Fig. S1C)” in the **RESULTS**.

- The paper demonstrates that a M77A mutation renders Irgm2 non-functional. The mutated methionine is part of the unconventional GMS P-loop sequence of IRGM proteins suggesting that the alanine mutations disrupts nucleotide (GDP) binding - see Hunn et al. (2008). Ideally, the M77A mutant should be biochemically characterized. Regardless, the authors should be more careful in their description of the mutant and distinguish between GTPase activity and nucleotide binding

We are grateful to the reviewer’s insight and apologize the misunderstanding of GMS p-loop of IRGM. In the revised manuscript, we describe the GMS p-loop sequence of Irgm2 as follows;

“Irgm2 harbors GMS sequence in the GTPase domain, in which the mutated methionine is a part of the unconventional GMS P-loop sequence of IRGM proteins (Bekpen et al., 2005). Therefore, we examined the role of the Irgm2 GTPase domain. A point mutant, in which the methionine at the 77 position in the GTPase domain of Irgm2 was substituted with alanine (Irgm2 M77A), may disrupt nucleotide (GDP) binding and hence render non-functional (Hunn et al., 2008).” in the **RESULTS**.

- Although reasonably well written, the paper needs to be edited for language here and there - to provide one example from page 6: "We confirmed that the defects in Irgm2-deficient cells contained effector IRGs and GBPs at levels similar to wild-type cells (Fig. 1D)." Based on the data shown in Fig. 1D, I believe the authors are trying to make the point here that IRG and GBP effector proteins are expressed at comparable levels in WT and Irgm2 KO cells butt this information is not conveyed by the sentence.

We apologize for any mistakes in editing. The revised manuscript is carefully read by English native users and corrected in accordance with their proofreading. Based upon **Fig 1D**, as the reviewer suggested, we just tried to report that “the protein expression levels” are comparable at time points tested between WT and Irgm2-KO MEFs. Therefore, we amend the sentence as follows;

“We confirmed that IRG and GBP effector proteins are expressed at comparable levels in wild-type and Irgm2-deficient cells (Fig. 1D).” in the **RESULTS**.

- the tandem IRG Irgb1-b2 has indeed been shown to act as a decoy for secreted Toxoplasma ROP5/18 kinase complex (Lilue et al., 2013). However, this evolutionary adaptation to overcome microbial immune evasion is not the primary function of these proteins, which are believed to mainly provide host resistance by directly targeting microbes or microbe-containing vacuoles for destruction. Without any discussion of the literature the 'decoy' designation is confusing. I'd suggest to either refer to GKS proteins simply as 'effectors,' or alternatively provide some more background, so that the non-expert reader understands why IRGs are here being referred to as both effectors and decoys. Please, cite Lilue et al., 2013 PMID: 24175088

We agree with the reviewer’s recommendation, delete “decoy” and simply divide IRG into “effectors” and “regulators” in the **ABSTRACT** and **INTRODUCTION**.

- The review article on IRGs published by Martens and Howard in 2006 is a bit dated - I'd suggest to also cite a more recent review by Pilla-Moffett et al. (2016) PMID: 27181197

We thank the reviewer’s suggestion and add the citation in the **INTRODUCTION** as follows;.

“In mice, the IRG family protein consists of three regulator proteins (Irgm1, Irgm2, Irgm3) and over 20 effector proteins (Bekpen et al., 2005; Pilla-Moffett et al., 2016).”

- Please cite Mukhopadhyay et al. (2020) PMID: 32293748 which made observations similar to Haldar et al. (2015) by demonstrating a role for TRAF6 (and TRAF2) in PVM ubiquitination and TRAF6-dependent killing of Toxo in IFN γ -primed mouse fibroblasts

We add the citation in the **INTRODUCTION** as follows;

“A subsequent recent study demonstrate a role for TRAF6 (and TRAF2) in PVM ubiquitination and TRAF6-dependent killing of *T. gondii* in IFN- γ -primed mouse fibroblasts (Mukhopadhyay et al., 2020).” in the **INTRODUCTION**.

- The Irgm1/m3-mediated protection of host endomembranes was demonstrated by Haldar et al. (2013) PMID: 23785284 - please cite in introduction when protection of endomembranes is discussed

We add the citation in the **INTRODUCTION** as follows;

“The negative regulation might be potentially important for protection of host endomembranes, since Irgm1 and Irgm3 are shown to localize at host Golgi apparatus and endoplasmic reticulum (ER), respectively (Martens et al., 2004; Hunn et al., 2008; Haldar et al., 2013).”

- Please also cite Park et al. (2016) PMID: 27172324 for the role the ATG system plays in distributing IRGs and GBPs

We add the citation in the **INTRODUCTION** as follows;

“In addition, the localization of IRGs and GBPs are also regulated by autophagy related proteins (ATGs) in a manner independent on autophagy (Zhao *et al.*, 2008; Yamamoto *et al.*, 2012; Ohshima *et al.*, 2014; Park *et al.*, 2016; Sasai *et al.*, 2017).” in the **INTRODUCTION**.

- *Two recently published papers reported 2 novel Irgm2 KO mouse models and described a role for Irgm2 in controlling LPS-mediated caspase-11 activation (Finethy *et al.* PMID: 33124745 and Eren *et al.* PMID: 33124769). It seems to make sense for the authors to discuss and contrast their findings with these recent reports.*

We newly discuss about the role of caspase-11 in the high mortality observed in Irgm2 KO mice with the requested citations in the **DISCUSSION** as follows;

“Moreover, recent studies demonstrate a role for Irgm2 in controlling LPS-mediated caspase-11 activation (Eren *et al.*, 2020; Finethy *et al.*, 2020). Although the role of caspase-11 in *T. gondii* infection in vivo remains to be seen, dysregulated caspase-11 activation as well as defective membrane trafficking might affect the high mortality in Irgm2-deficient mice in *T. gondii* infection.” in the **DISCUSSION**.

- *There is no description of the TKO cells/ mouse in Materials and Methods - please, provide this information*

We thank the reviewer’s suggestion and include the information about the generation of Irgm1/m2/m3 TKO mice in the **MATERIALS AND METHODS** with the protein expression data (**Fig. S6B**) as follows;

“Irgm1/Irgm2/Irgm3-TKO mice were generated through the genome editing by introducing the Irgm2 targeting gRNAs together with the Irgm1- or Irgm3-targeting gRNAs into mouse embryos (Lee *et al.*, 2015). The Irgm1/Irgm2/Irgm3-TKO mice were born at Mendelian ratios and were healthy. The expression of the proteins for Irgm1, Irgm2 and Irgm3 in primary

embryonic fibroblasts was analyzed by Western blotting (**Fig. S6B**)." in the **MATERIALS AND METHODS**.

- Fig. 5A-D lack controls: i.e. WT cells and cells deficient for Ub loading (e.g. Atg3 KO or Irgm1/m3 KOs that were used in a previous publication from the lab by Lee et al. (2015))

In response to the reviewer's request, we newly compared the Ub loading as well as recruitment of Gbp1, Irgb6 and p62 among wild-type, Irgm2 KO and Irgm1/m3 DKO cells. As reported previously (Lee et al. Cell Rep. 2015), the Ub loading and recruitment of these effectors were greatly reduced in Irgm1/m3 DKO cells. Compared with Irgm1/m3 DKO cells, Irgm2-KO cells were partially defective in loading of these effectors. The data are added in **Fig. S3B** with sentences as follows;

"When we compared the ubiquitin loading as well as recruitment of Gbp1, Irgb6 and p62 among wild-type, Irgm2-deficient and Irgm1/m3 DKO cells, we found that the ubiquitin loading and recruitment of these effectors were greatly reduced in Irgm1/m3 DKO cells as reported previously (Lee *et al.*, 2015). Compared with Irgm1/m3 DKO cells, Irgm2-deficient cells were partially defective in loading of these effectors (**Fig. S3B**)." in the **RESULTS**.

- Figure 6C: The authors should provide the p-value of the comparison of empty vector/+gamma vs KA/+ gamma

We thank the reviewer's suggestion. To respond to a major comment of the Reviewer #1, we have changed the statistical methods throughout figures. For **Fig. 6C**, the P value of empty vector/+ gamma is 0.0006 (2 way ANOVA, Tukey's multiple comparisons test).

March 27, 2021

Re: Life Science Alliance manuscript #LSA-2020-00960-TR

Prof. Masahiro Yamamoto
Research Institute for Microbial Diseases
Osaka University
3-1, Yamadaoka
Suita city, Osaka 565-0871
Japan

Dear Dr. Yamamoto,

Thank you for submitting your manuscript entitled "Cell-autonomous Toxoplasma killing program requires Irgm2 but not its microbe vacuolar localization" to Life Science Alliance. The manuscript was assessed by expert reviewers, whose comments are appended to this letter.

We apologize for this extended and unusual delay in getting back to you. As you will note from the reviewers' comments below, one of the reviewers' (Reviewer 2 below & Reviewer 3 at the previous journal) is still dissatisfied with the revised manuscript, citing that a number of conclusions are not supported by data, some experiments lack of scientific rigor and that the very low IFN γ concentrations seen are anything other than noise. We agree with the reviewers' concerns. The reviewer has provided a roadmap to revise the manuscript, and we would encourage you to submit a revised version back to us that addresses the reviewer 2's points.

Thank you for this interesting contribution to Life Science Alliance. We are looking forward to receiving your revised manuscript.

Sincerely,

Shachi Bhatt, Ph.D.

Executive Editor

Life Science Alliance

<https://www.lsjournal.org/>

Interested in an editorial career? EMBO Solutions is hiring a Scientific Editor to join the international Life Science Alliance team. Find out more here -

https://www.embo.org/documents/jobs/Vacancy_Notice_Scientific_editor_LSA.pdf

B. MANUSCRIPT ORGANIZATION AND FORMATTING:

Reviewer #1 (Comments to the Authors (Required)):

In murine cells the cytokine interferon gamma upregulates IRGs and GBPs, which play an important

role in the destruction of the parasitophorous vacuole. Therefore, understanding the exact mechanism by which these IRGs and GBPs are recruited to the PVM is important.

In this manuscript the authors show that *Irgm2* deficiency leads to impaired *T. gondii* killing activity in vitro with selectively decreased recruitment of *Irgb6* and *Gbp1* on the PVM. *Irgm2*^{-/-} mice are very susceptible to *Toxoplasma* demonstrating its in vivo importance. The GMS configuration of *Irgm2* is essential for the cell-autonomous immune function of *Irgm2*. IRGs, Atg, GBP1, *Irgb6* proteins are not involved in *Irgm2* localization on *T. gondii* PVM. Cys358 of *Irgm2* is a determinant for the localization on the *T. gondii* PVM and for its localization at the Golgi apparatus in uninfected cells. However, neither PVM localization nor Golgi localization is needed for its function as *Toxoplasma* growth inhibition is restored by this mutant as is GBP1/*Irgb6* PVM localization. However, prolonged (8hrs but not 4hrs) accumulation of p62 and ubiquitin is dependent on *Irgm2* vacuole localization and presentation of vacuolar antigen to activate CD8⁺ T cells. *Irgm2* is ubiquitinated and ubiquitinated *Irgm2* interacts with *Gbp1* to promote recruitment of *Gbp1* to *T. gondii* PVM.

Overall their data strongly supports their model.

In this revised version they have made it more clear that cytosolic ubiquitinated *Irgm2* is required for proper localization of *Gbp1* to the PVM and that it is likely that the absence of *Gbp1* at the PVM is responsible for some of the phenotypes observed in their *Irgm2* mutant expressing cells. This takes away the confusion that arose with the explanation of their data in the first version of this manuscript.

Reviewer #2 (Comments to the Authors (Required)):

Unfortunately, I don't find the revised manuscript to be substantially improved. Critical technical concerns have not been addressed and new concerns regarding scientific rigor have emerged. While I appreciate that the authors have followed the editors' recommendation and 'toned down' their claim that *Irgm2* plays a role in antigen presentation, the authors failed to address the technical concerns I had raised regarding the single piece of data purportedly monitoring antigen presentation. Moreover, new data provided in the revised manuscript directly contradicts findings published by the Yamamoto lab in the past raising substantial concerns regarding scientific rigor. Additionally, the authors are now emphasizing claims of causality (in the abstract and elsewhere) that are only based on correlation and unlikely to be true. And last but not least, the author make a new claim ("*Irgm2* itself is not an ubiquitin substrate at the PVM") that is not supported by their data and unlikely to be true. Overall, I have some concerns regarding scientific rigor and substantial concerns regarding data interpretation. However, all of my concerns are addressable. To do so will require additional experimental work and, maybe more importantly, careful (re)consideration of data interpretation and substantial text editing.

Major

- The authors provide new data comparing parasitic burden in *Irgm2* KO, *Irgm1/m3* DKO, and *Irgm1/m2/m3* TKO MEFs (new Fig. S1C) and show that *Irgm1/m3* DKO have a partial defect in host resistance. This finding is in contrast to their previously published data (Lee et al. (2015) PMID: 26440898, Fig. S3M), in which the Yamamoto lab showed that *Irgm1/m3* DKO MEFs displayed a complete loss in IFN γ -inducible resistance to *Toxoplasma* infections. Which report is accurate? The data from Lee et al. 2015 or the data shown here? It seems that the exact same cells, *Toxoplasma* strain and assays system were used. This needs to be addressed carefully, potentially through the use of alternative assays other than the luciferase system. If the authors conclude that *Irgm1/m3* DKO MEFs have indeed an intermediate phenotype, they should provide an

explanation for the contradictory findings reported by their own group previously (Lee et al., 2015)

- The revised abstract fails to make sense - here's why: it raises an interesting question "very little is known about the significance and mechanism of the unique localization [of Irgm2] on Toxoplasma PVM" and then ends in "taken together, these data indicate the IFN-inducible GTPase-dependent cell-autonomous immunity is controlled in a manner dependent or independent on the Irgm2 localization at the Toxoplasma PVM." In other words, the abstract raises two questions, ignores one of the questions entirely (mechanism - how does Irgm2 get to the PVM?) and concludes that it doesn't have an answer to the second question (significance - what does Irgm2 do at the PVM anyways?)

- In my previous critique I raised the point that "an Irgm2 mutant lacking all lysines (Irgm2-KA) still translocates to the Toxo PV but fails to restore complete PV ubiquitination in Irgm2 KO cells. Therefore, the partial drop in PV ubiquitination seen in Irgm2 KO cells appears to be simply due to the absence of ubiquitinated Irgm2 from the PV. In other words, Irgm2-dependent PV ubiquitination is simply the accumulation of ubiquitinated Irgm2 at the PV." In the revised MS (and the rebuttal) the authors argue that "on the other hand, the Irgm2 KA mutant-reconstituted Irgm1/Irgm2/Irgm3 TKO cells exhibited recruitment of the KA mutant as comparable as wild-type Irgm2 but showed no ubiquitination on PVM, suggesting that Irgm2 itself is not an ubiquitin substrate at the PVM." This conclusion is not justified and the argument is not logical. Here's why. The authors had already established that the KA mutant translocates to the PVM and therefore we know that ubiquitination of Irgm2 is not required for its translocation to the PVM. The authors had also already established that the system responsible for attachment of ubiquitin to PVM-resident proteins or to proteins destined to be delivered to the PVM is broken in Irgm1/Irgm2/Irgm3 TKO cells. Therefore, we learned nothing new from expressing the KA mutant or WT Irgm2 in Irgm1/Irgm2/Irgm3 TKO cells - Irgm2 KA can still translocate to PVMs, as expected, and Irgm1/Irgm2/Irgm3 TKO cells are still deficient for PVM ubiquitination, as expected. In order to draw any meaningful conclusions, the authors need to directly monitor the ubiquitination status of PVM-resident Irgm2 in WT cells. There are many ways to do this, including biochemical approaches or proximity labeling. This is certainly the type of experiment that the Yamamoto lab is equipped to conduct

- In the revised abstract and throughout the MS the authors claim that "Irgm2 [...] is important for parasite killing through the recruitment of Gbp1 to the PVM." This statement is not supported by the data and, based on what we do know, very unlikely to be correct. First of all, the authors take a correlation (reduced translocation of Gbp1 to PVMs in Irgm2 KO cells correlates with reduced parasite killing in Irgm2 KO cells) and turn it into causation (reduced translocation of Gbp1 to PVMs in Irgm2 KO cells causes reduced parasite killing in Irgm2 KO cells) without additional evidence of causality. Furthermore, the partial defect in Gbp1 recruitment is unlikely to explain the pronounced phenotypes of Irgm2 KO cells and animals (in other words, the correlation isn't even that good). And thirdly, Gbp1 KO mice have a relatively moderate increase in susceptibility to Toxoplasma (Selleck et al. 2013 PMID: 23633952), not resembling the severe immune defect of Irgm2 KO mice. The authors should edit the text throughout and provide a better discussion of potential mechanisms by which Irgm2 may mediate host defense to Toxoplasma. Maybe they don't really have a good explanation - that's fine but then they should just say so.

- The authors have not addressed my concerns regarding the data depicted in Fig. 5E - from my previous critique: "1) the IFN γ concentrations are exceedingly low (25 ng/ml), i.e. barely above the level of cytokines produced by T cells co-cultured with MEF / Toxo parental controls - compare this to Fig. 4H in Lee et al (2015) where the same assay was run by the Yamamoto lab and IFN γ concentrations were more than 1 log higher" - I appreciate that the authors have "toned down" their claims regarding the role for Irgm2 in antigen presentation. However, I maintain that the data shown here are not of sufficient scientific rigor: it is unclear whether these very low IFN γ concentrations are anything other than noise. This is of particular concern since there are no alternative approaches provided to monitor antigen presentation. The authors should either

conduct additional experiments or remove these data entirely.

Minor

- If the authors decide to keep their "antigen presentation data" in this study, they should also address at least one other point from my previous critique, namely "3) MEFs are not professional APCs and not of relevance for Toxo in vivo infections - the study should determine whether Irgm2 KO DCs have a defect in antigen presentation (as done by the group in Lee et al. in the analysis of p62 KO and Irgm1/m3 KO DCs)"
- Were the studies conducted with a single MEF line per genotype (derived from one embryo) or three or more biologically independent MEF lines per genotype (biological replicates)? This is related to one of the major point raised, namely that Irgm1/m3 DKO MEFs used in this study display a different phenotype than what the group reported in Lee et al. Under Materials & Methods the authors should specify how their MEFs were made and whether multiple independent embryos were used for each genotype
- Please provide the sgRNA sequences targeting Irgm2 used for the production of the Irgm1/m2/m3 TKO mouse; this is standard information provided for any CRISPR generated mice or cell line
- The new Fig S3B lists 4 genotypes in the chart legend but only shows data for 3 genotypes
- As stated in my previous critique, it seems unclear to me what model the authors are proposing for Irgm2's purported function - does Irgm2 execute a defense mechanism that is distinct from the one controlled by Irgm1/m3? The basic outline of their model should be discussed
- Although the rebuttal states that "the revised manuscript is carefully read by English native users and corrected in accordance with their proofreading," I don't see any evidence of that. The new text added to the revised MS is also riddled with errors

Point-by-Point Responses to Reviewer #2:

Unfortunately, I don't find the revised manuscript to be substantially improved. Critical technical concerns have not been addressed and new concerns regarding scientific rigor have emerged. While I appreciate that the authors have followed the editors' recommendation and 'toned down' their claim that Irgm2 plays a role in antigen presentation, the authors failed to address the technical concerns I had raised regarding the single piece of data purportedly monitoring antigen presentation. Moreover, new data provided in the revised manuscript directly contradicts findings published by the Yamamoto lab in the past raising substantial concerns regarding scientific rigor. Additionally, the authors are now emphasizing claims of causality (in the abstract and elsewhere) that are only based on correlation and unlikely to be true. And last but not least, the author make a new claim ("Irgm2 itself is not an ubiquitin substrate at the PVM") that is not supported by their data and unlikely to be true. Overall, I have some concerns regarding scientific rigor and substantial concerns regarding data interpretation. However, all of my concerns are addressable. To do so will require additional experimental work and, maybe more importantly, careful (re)consideration of data interpretation and substantial text editing.

We thank the reviewer's comments and careful reading.

Major

- The authors provide new data comparing parasitic burden in Irgm2 KO, Irgm1/m3 DKO, and Irgm1/m2/m3 TKO MEFs (new Fig. S1C) and show that Irgm1/m3 DKO have a partial defect in host resistance. This finding is in contrast to their previously published data (Lee et al. (2015) PMID: 26440898, Fig. S3M), in which the Yamamoto lab showed that Irgm1/m3 DKO MEFs displayed a complete loss in IFN γ -inducible resistance to Toxoplasma infections. Which report is accurate? The data from Lee et al. 2015 or the data shown here? It seems that the exact same cells, Toxoplasma strain and assays system were used. This needs to be addressed carefully, potentially through the use of alternative assays other than the luciferase system. If the authors conclude that Irgm1/m3 DKO MEFs have indeed an intermediate phenotype,

they should provide an explanation for the contradictory findings reported by their own group previously (Lee et al., 2015)

Both are accurate. Depending on various conditions of parasite growth and host cells, it is possible that the relative luciferase expression levels showed more than 100% (See the lanes of red and green of IFN- γ -stimulated conditions in Fig. 2C), meaning that parasites in the IFN- γ -stimulated MEFs grew more than those in unstimulated cells. As of 2015, although we did not have Irgm1/Irgm2/Irgm3 TKO cells, if we had and used them, Irgm1/Irgm2/Irgm3 TKO cells could have shown severer phenotype than Irgm1/Irgm3 DKO cells in Lee et al (2015). The data in Lee et al (2015) can say that Irgm1/Irgm3 DKO cells are defective in IFN- γ -induced reduction of *T. gondii* numbers in comparison to wild-type cells. The data in Fig. S1C in the present study can say that Irgm1/Irgm2/Irgm3 TKO cells showed more severe phenotype than Irgm1/Irgm3 DKO cells or just that "However, the magnitude of the defect in Irgm2-single deficient cells was not as severe as Irgm1/Irgm3 double KO (DKO) or Irgm1/Irgm2/Irgm3 triple knockout (TKO) cells (Fig. 1A and Fig. S1C)". Thus, we can only compare the relative severity of each genotype in the same assay but not the severity itself. Therefore, both findings are not contradicted at all.

- The revised abstract fails to make sense - here's why: it raises an interesting question "very little is known about the significance and mechanism of the unique localization [of Irgm2] on Toxoplasma PVM" and then ends in "taken together, these data indicate the IFN-inducible GTPase-dependent cell-autonomous immunity is controlled in a manner dependent or independent on the Irgm2 localization at the Toxoplasma PVM." In other words, the abstract raises two questions, ignores one of the questions entirely (mechanism - how does Irgm2 get to the PVM?) and concludes that it doesn't have an answer to the second question (significance - what does Irgm2 do at the PVM anyways?)

We agree with the reviewer's comment. The biological significance of the Irgm2 localization on the PVM is unknown after the analysis since we decide to remove the antigen presentation data in accordance with the reviewer's strong recommendation (See below). Therefore, we have emphasized that the significance of Irgm2 localization on *T. gondii* PVM is still unclear in the Discussion of the revised manuscript as follows;

"We could not dissect the biological significance of Irgm2 localization on the *T. gondii* PVM in this study. Because Irgm2 localization on the *T. gondii* PVM was required for prolonged recruitment of p62 and ubiquitin to the vacuole without affecting parasite killing, it would be of interest to examine the role of Irgm2 localization on the *T. gondii* PVM in parasite killing-independent anti-*T. gondii* responses in the future." in the **Discussion.**

- In my previous critique I raised the point that "an Irgm2 mutant lacking all lysines (Irgm2-KA) still translocates to the Toxo PV but fails to restore complete PV ubiquitination in Irgm2 KO cells. Therefore, the partial drop in PV ubiquitination seen in Irgm2 KO cells appears to be simply due to the absence of ubiquitinated Irgm2 from the PV. In other words, Irgm2-dependent PV ubiquitination is simply the accumulation of ubiquitinated Irgm2 at the PV." In the revised MS (and the rebuttal) the authors argue that "on the other hand, the Irgm2 KA mutant-reconstituted Irgm1/Irgm2/Irgm3 TKO cells exhibited recruitment of the KA mutant as comparable as wild-type Irgm2 but showed no ubiquitination on PVM, suggesting that Irgm2 itself is not an ubiquitin substrate at the PVM." This conclusion is not justified and the argument is not logical. Here's why. The authors had already established that the KA mutant translocates to the PVM and therefore we know that ubiquitination of Irgm2 is not required for its translocation to the PVM. The authors had also already established that the system responsible for attachment of ubiquitin to PVM-resident proteins or to proteins destined to be delivered to the PVM is broken in Irgm1/Irgm2/Irgm3 TKO cells. Therefore, we learned nothing new from expressing the KA mutant or WT Irgm2 in Irgm1/Irgm2/Irgm3 TKO cells - Irgm2 KA can still translocate to PVMs, as expected, and Irgm1/Irgm2/Irgm3 TKO cells are still deficient for PVM ubiquitination, as expected. In order to draw any meaningful conclusions, the authors need to directly monitor the ubiquitination status of PVM-resident Irgm2 in WT cells. There are many ways to do this, including biochemical approaches or proximity labeling. This is certainly the type of experiment that the Yamamoto lab is equipped to conduct

To respond to the reviewer's request biochemically, the purification of Irgm2 proteins only on *Toxoplasma* PVM is essential. However, such methods to isolate *Toxoplasma* PVM have not been established yet, therefore, undoable. In addition, although the reviewer requests proximity labeling in the other option,

proximity labelling cannot prove the direct interaction of (direct ubiquitination in this case) the protein of interest. Various host proteins such as ubiquitin, p62, effector IRGs and GBPs in addition to Irgm2 are recruited on *Toxoplasma* PVM (Saeij and Frickel 2017 PMID: 29141239). In such situations where ubiquitin and Irgm2 localize at the same place, the proximity labeling cannot distinguish the direct labeling or indirect labeling between ubiquitin and Irgm2 even if the proximity labeling signals could be detected. Thus, both biochemical approaches and proximity labeling to answer the reviewer's question are challenging (out of scope of the current study) and meaningless, respectively.

*- In the revised abstract and throughout the MS the authors claim that "Irgm2 [...] is important for parasite killing through the recruitment of Gbp1 to the PVM." This statement is not supported by the data and, based on what we do know, very unlikely to be correct. First of all, the authors take a correlation (reduced translocation of Gbp1 to PVMs in Irgm2 KO cells correlates with reduced parasite killing in Irgm2 KO cells) and turn it into causation (reduced translocation of Gbp1 to PVMs in Irgm2 KO cells causes reduced parasite killing in Irgm2 KO cells) without additional evidence of causality. Furthermore, the partial defect in Gbp1 recruitment is unlikely to explain the pronounced phenotypes of Irgm2 KO cells and animals (in other words, the correlation isn't even that good). And thirdly, Gbp1 KO mice have a relatively moderate increase in susceptibility to *Toxoplasma* (Selleck et al. 2013 PMID: 23633952), not resembling the severe immune defect of Irgm2 KO mice. The authors should edit the text throughout and provide a better discussion of potential mechanisms by which Irgm2 may mediate host defense to *Toxoplasma*. Maybe they don't really have a good explanation - that's fine but then they should just say so.*

Any previous publications regarding the role of IRGs, GBPs and effectors in anti- *Toxoplasma* host defense only showed correlation and connected reduced effector localization on *T. gondii* PVM in vitro with the in vivo phenotype as the causation (Selleck et al. 2013 PMID: 23633952; Degrandi et al. 2013 PMID: 23248289; Steffens et al. 2020 PMID: 31964735; Zhao et al. 2009 PMID: 19265156; Choi et al. 2012 PMID: 24931121; Zhao et al. 2008 PMID: 18996346). Therefore, the current way to interpret the data for Irgm2 in this manuscript is followed by the standard but not unusual at all. However, we agree with the reviewer's comment that all of the previous papers including our

current manuscript cannot strictly connect the correlation and the causation. We discuss the point as follows;

“Compared with other regulatory IRGs, considering that the time course of death of Irgm2-deficient mice after *T. gondii* infection was similar to that of Irgm3/iNOS double-deficient mice (Zhao *et al.*, 2009), Irgm2-deficient mice might be more susceptible to *T. gondii* than Irgm3-deficient mice *in vivo*. We found that Irgm2 deficiency led to partially defective translocation of Gbp1 and Irgb6 or reduced retention of p62 and ubiquitin on the *T. gondii* PVM. However, considering that Gbp1-deficient mice have a relatively moderate increase in their *in vivo* susceptibility to *T. gondii* (Selleck *et al.* 2013), the partial defects in Gbp1 recruitment to the *T. gondii* PVM by Irgm2 deficiency might not be sufficient to explain the severe *in vivo* phenotype of Irgm2-deficient mice. Although additional defects in the recruitment of Irgb6, p62, and ubiquitin to the *T. gondii* PVM may account for the severe phenotype of Irgm2-deficient mice, whether the correlative reduction of IFN-inducible effector recruitment to the *T. gondii* PVM could strictly connect with the causation of the *in vivo* phenotype should be carefully examined in the future.” in the **Discussion**.

*- The authors have not addressed my concerns regarding the data depicted in Fig. 5E - from my previous critique: "1) the IFN γ concentrations are exceedingly low (25 ng/ml), i.e. barely above the level of cytokines produced by T cells co-cultured with MEF / Toxo parental controls - compare this to Fig. 4H in Lee *et al* (2015) where the same assay was run by the Yamamoto lab and IFN γ concentrations were more than 1 log higher" - I appreciate that the authors have "toned down" their claims regarding the role for Irgm2 in antigen presentation. However, I maintain that the data shown here are not of sufficient scientific rigor: it is unclear whether these very low IFN γ concentrations anything other than noise. This is of particular concern since there are no alternative approaches provided to monitor antigen presentation. The authors should either conduct additional experiments or remove these data entirely.*

In accordance with the reviewer's recommendation, we have removed the antigen presentation data entirely in this revised manuscript.

Minor

- If the authors decide to keep their "antigen presentation data" in this study, they should also address at least one other point from my previous critique,

namely "3) MEFs are not professional APCs and not of relevance for Toxo in vivo infections - the study should determine whether Irgm2 KO DCs have a defect in antigen presentation (as done by the group in Lee et al. in the analysis of p62 KO and Irgm1/m3 KO DCs)"

Since we have removed the antigen presentation data, the experiments using Irgm2 KO DCs are not required anymore.

- Were the studies conducted with a single MEF line per genotype (derived from one embryo) or three or more biologically independent MEF lines per genotype (biological replicates)? This is related to one of the major point raised, namely that Irgm1/m3 DKO MEFs used in this study display a different phenotype than what the group reported in Lee et al. Under Materials & Methods the authors should specify how their MEFs were made and whether multiple independent embryos were used for each genotype

We established two independent Irgm2 KO MEFs from two embryos with similar defects in anti- *Toxoplasma* responses (**Supporting Fig. 1A and 1B**).

[Figure removed by editorial staff per authors' request]

Supporting Figure 1 Two lines of Irgm2 KO MEFs showing the similar defective anti-*T. gondii* response

(A) Western blot image of indicated protein expression in WT and Irgm2 KO MEFs (#1 and #2) at indicated hours post IFN- γ activation.

(B) Comparison of *T. gondii* survival rate in indicated MEFs with IFN- γ stimulation relative to those without IFN- γ treatment by luciferase analysis at 24 h post infection.

Difference in *T. gondii* inhibition activity between IFN- γ activated vs non-activated was subjected to two-way ANOVA, with Tukey's multiple comparisons test to analyze the difference between genotypes.

We have described the fact in the Materials and Methods as follows;

“Two embryos were used to generate two independent Irgm2-deficient MEF lines that similarly showed defects in anti-*T. gondii* responses (data not shown).” in the **Materials and Methods.**

- Please provide the sgRNA sequences targeting Irgm2 used for the production of the Irgm1/m2/m3 TKO mouse; this is standard information provided for any CRISPR generated mice or cell line

The sgRNA sequence targeting Irgm2 is provided in the Materials and Methods as follows;

“The target gRNA sequence of Irgm2 was 5'-GAGAAAGATTTCAGCTCCCACTGG-3' (TGG; the PAM sequence) in the N-terminus.” and “Irgm1/Irgm2/Irgm3-TKO mice were generated through genome editing by introducing the same Irgm2-targeting gRNA used to generate Irgm2-deficient mice as described above together with Irgm1- or Irgm3-targeting gRNAs into mouse embryos (Lee *et al.*, 2015).” in the **Materials and Methods.**

- The new Fig S3B lists 4 genotypes in the chart legend but only shows data for 3 genotypes

We have corrected the list.

- As stated in my previous critique, it seems unclear to me what model the authors are proposing for Irgm2's purported function - does Irgm2 execute a defense mechanism that is distinct from the one controlled by Irgm1/m3? The basic outline of their model should be discussed

Irgm1 and Irgm3 globally control anti- *Toxoplasma* defense mechanism involving Irga6, Irgb6, Gbp1 and Gbp2. On the other hand, Irgm2 specifically involves the host defense involving Irgb6 and Gbp1 but not Irga6 and Gbp2. This is the proposed function of Irgm2 described in this manuscript. The point is now described in the Discussion as follows;

“Irgm2-deficient cells showed specific defects in recruitment of Irgb6 and Gbp1 but not Irga6 or Gbp2, which suggests selective requirement of Irgm2 in Irgb6- and Gbp1-dependent anti-*T. gondii* responses. Conversely, Irgm1 and Irgm3 globally control the anti-*T. gondii* cellular response that involves Irga6, Irgb6, Gbp1, and Gbp2

(Lee et al. 2015). Thus, Irgm2 uniquely participates in IFN-inducible GTPase-mediated cell-autonomous immunity against *T. gondii*." in the **Discussion**.

- Although the rebuttal states that "the revised manuscript is carefully read by English native users and corrected in accordance with their proofreading," I don't see any evidence of that. The new text added to the revised MS is also riddled with errors

We have outsourced English editing in the revised manuscript throughout the text. The evidence of proofreading by attaching the modification record is attached as the supplemental document.

May 10, 2021

RE: Life Science Alliance Manuscript #LSA-2020-00960-TRR

Prof. Masahiro Yamamoto
Research Institute for Microbial Diseases
Osaka University
3-1, Yamadaoka
Suita city, Osaka 565-0871
Japan

Dear Dr. Yamamoto,

Thank you for submitting your revised manuscript entitled "Cell-autonomous Toxoplasma killing program requires Irgm2 but not its microbe vacuolar localization". We would be happy to publish your paper in Life Science Alliance pending final revisions necessary to meet our formatting guidelines.

Along with the points listed below, please also attend to the following:

- please add an Author Contributions section to your main manuscript text
- please revise the legend for figures 6 (missing H in legend for Figure 6) and S5 so that the panels are introduced in order
- please provide higher quality images of western blots in Figures 1D, 2B, 6B, I

A. FINAL FILES:

- An editable version of the final text (.DOC or .DOCX) is needed for copyediting (no PDFs).
- High-resolution figure, supplementary figure and video files uploaded as individual files: See our detailed guidelines for preparing your production-ready images, <https://www.life-science-alliance.org/authors>
- Summary blurb (enter in submission system): A short text summarizing in a single sentence the

study (max. 200 characters including spaces). This text is used in conjunction with the titles of papers, hence should be informative and complementary to the title. It should describe the context and significance of the findings for a general readership; it should be written in the present tense and refer to the work in the third person. Author names should not be mentioned.

B. MANUSCRIPT ORGANIZATION AND FORMATTING:

Sincerely,

Shachi Bhatt, Ph.D.
Executive Editor
Life Science Alliance
<http://www.lsajournal.org>
Tweet @SciBhatt @LSAJournal

May 20, 2021

RE: Life Science Alliance Manuscript #LSA-2020-00960-TRRR

Prof. Masahiro Yamamoto
Research Institute for Microbial Diseases
Osaka University
3-1, Yamadaoka
Suita city, Osaka 565-0871
Japan

Dear Dr. Yamamoto,

Thank you for submitting your Research Article entitled "Cell-autonomous Toxoplasma killing program requires Irgm2 but not its microbe vacuolar localization". It is a pleasure to let you know that your manuscript is now accepted for publication in Life Science Alliance. Congratulations on this interesting work.

DISTRIBUTION OF MATERIALS:

Again, congratulations on a very nice paper. I hope you found the review process to be constructive and are pleased with how the manuscript was handled editorially. We look forward to future exciting submissions from your lab.

Sincerely,

Shachi Bhatt, Ph.D.

Executive Editor

Life Science Alliance

<http://www.lsajournal.org>
